# FABP3-mediated membrane lipid saturation alters fluidity and induces ER stress in skeletal muscle with aging

Seung-Min Lee [1], Seol Hee Lee[1,2], Youngae Jung[3], Younglang Lee[4], Jong Hyun Yoon[1,5], Jeong Yi Choi[1], Chae Young Hwang[1], Young Hoon Son[1], Sung Sup Park[1,2], Geum-Sook Hwang[3], Kwang-Pyo Lee [1,2,4 ✉] & Ki-Sun Kwon [1,4,5 ✉]

Sarcopenia is characterized by decreased skeletal muscle mass and function with age. Aged muscles have altered lipid compositions; however, the role and regulation of lipids are unknown. Here we report that FABP3 is upregulated in aged skeletal muscles, disrupting homeostasis via lipid remodeling. Lipidomic analyses reveal that FABP3 overexpression in young muscles alters the membrane lipid composition to that of aged muscle by decreasing polyunsaturated phospholipid acyl chains, while increasing sphingomyelin and lysophosphatidylcholine. FABP3-dependent membrane lipid remodeling causes ER stress via the PERK-eIF2α pathway and inhibits protein synthesis, limiting muscle recovery after immobilization. FABP3 knockdown induces a young-like lipid composition in aged muscles, reduces ER stress, and improves protein synthesis and muscle recovery. Further, FABP3 reduces membrane fluidity and knockdown increases fluidity in vitro, potentially causing ER stress. Therefore, FABP3 drives membrane lipid composition-mediated ER stress to regulate muscle homeostasis during aging and is a valuable target for sarcopenia.

[1] Aging Research Center, Korea Research Institute of Bioscience and Biotechnology (KRIBB), Daejeon 34141, Republic of Korea. [2] Department of Biomolecular Science, KRIBB School of Bioscience, Korea University of Science and Technology, Daejeon 34113, Republic of Korea. [3] Integrated Metabolomics Research Group, Western Seoul Center, Korea Basic Science Institute, Seoul 03759, Republic of Korea. [4] Aventi Inc., Daejeon 34141, Republic of Korea. [5] Department of Functional Genomics, KRIBB School of Bioscience, Korea University of Science and Technology, Daejeon 34113, Republic of Korea. ✉email: kplee@kribb.re.kr; kwonks@kribb.re.kr

Aged skeletal muscle exhibits loss of muscle mass and function, defined as sarcopenia[1], contributing to a number of unfavorable consequences, including falls, fractures, frailty, and mortality in the elderly[2,3]. Owing to the importance of age-related muscle weakness for overall health of the elderly, the World Health Organization has established ICD-10-CM disease code for sarcopenia[4]. Sarcopenia can be triggered by a variety of conditions, including hormonal imbalances, loss of motor neurons, inflammation, defective mitochondrial homeostasis, and altered body composition[5]. In addition, intramuscular lipid and metabolite accumulation increases with age, possibly leading to skeletal muscle insulin resistance and systemic inflammation[6–8]. However, little is known about the underlying mechanism, key players, or downstream effectors of lipid remodeling dysfunction in skeletal muscles during aging.

Fatty acid binding proteins (FABPs) are 14–15 kDa proteins that bind to hydrophobic ligands such as long chain fatty acids[9]. FABPs include nine members with tissue-specific expression patterns[10]. Among FABPs, the role of FABP4 and FABP5 are relatively well-studied. FABP4/5 are highly expressed in adipose tissue and macrophages and contribute to metabolic diseases with adipose inflammation and insulin resistance[11–13]. In mice, Fabp4 deficiency in macrophages induces monounsaturated fatty acid production via stearoyl-CoA desaturase 1 (SCD-1) upregulation, which protects against atherosclerosis[14]. Moreover, Fabp4/5 double-deficient mice exhibit higher tolerance to high fat diet-induced obesity, insulin resistance, and diabetes[13]. FABP3 is highly expressed in the heart and skeletal muscles[15], and its expression in skeletal muscles is elevated on consumption of a high fat diet. Fabp3 partially deficient mice reportedly exhibit reduced fatty acid utilization[16]. FABP3 is considered to serve as a lipid "chaperone" regulating solubility, mobility, and utilization of fatty acids[10]. In our previous study, proteomic analysis identified that FABP3 is highly expressed in aged muscle[17]. However, how FABP3 contributes to muscle aging has not been addressed.

Here, we focus on elucidating age-associated FABP3 function as a lipid chaperone in skeletal muscle. Using comparative lipidomic analyses in young versus aged muscle, and in FABP3-overexpressing or knockdown muscle, we identify a unique age-associated lipid composition signature, which is altered by FABP3 abundance. Using fluorescence recovery after photobleaching (FRAP) analysis for myotubes, we investigate whether lipid composition affects membrane fluidity. We further evaluate FABP3-mediated downstream signaling and assesse muscle mass and contractility allowing us to propose a mechanism for skeletal muscle aging.

## Results

**Lipidomic signature of acyl chains saturation in aged skeletal muscle.** FABP3 expression was dramatically increased in skeletal muscle but not in cardiac muscle with age (Fig. 1a). Since FABP3 has been reported as a lipid "chaperone" which regulates fatty acid metabolism[10], we first investigated a lipidomic signature in "young" (3-month-old) and "aged" (24-month-old) mouse tibialis anterior (TA) muscle using UPLC/QTOF mass spectrometry. Our analysis identified 203 lipid species. Approximately 53% of total identified lipid species were significantly changed, with 45 lipid species increased and 62 decreased (Fig. 1b). We next analyzed the distribution and alteration of the lipid classes. The major lipid species identified in skeletal muscle were the membrane lipids phosphatidylcholine (PC, 75% of total identified lipid), phosphatidylethanolamine (PE, 11%), lysophosphatidylcholine (LPC), sphingomyelin (SM), and phosphatidylinositol (PI) (Supplementary Table 1). Aged muscle showed marked changes in the content of lipid classes (Supplementary Table 1 and

Supplementary Fig. 1a). Notably, aged muscle had elevated SM and LPC, showing 2.4-fold and 2.8-fold increases over young muscle, respectively (Fig. 1c and Supplementary Fig. 1a). We next analyzed acyl chain composition of detected PC, PE, SM, and LPC in aged muscle compared to young muscle. In aged muscle, 16:0/20:4 PC significantly decreased by 24% and 16:0/16:0 PC increased 2.1-fold over young muscle. Other PC species changed slightly or insignificantly (Fig. 1d and Supplementary Fig. 1b). When individual contents of acyl chains were summed, polyunsaturated PC acyl chains decreased in aged muscle, but saturated species increased (Fig. 1h and Supplementary Fig. 1c). The PE species containing 40:6, 38:4, and 40:8 acyl chains decreased in aged muscle by 45, 42, and 48%, respectively. However, 40:4, 34:1, and 36:1 PE increased more than 2-fold (Fig. 1e and Supplementary Fig. 1d). Identification of individual acyl chains was not possible for all lipid species, due high isomeric or isobaric lipid levels. When PE species containing the same number of double bonds were combined, PEs with more double bonds decreased in aged muscle, but PEs containing fewer double bonds increased (Supplementary Fig. 1e). These data indicate that PC and PE acyl chains were shifted from polyunsaturated to saturated in aged muscle. Interestingly, all identified SM and LPC species (which mainly contain saturated or monounsaturated acyl chains[18]) increased simultaneously in aged muscle (Fig. 1f, g and Supplementary Fig. 1f, g). In parallel, we evaluated PC acyl chain length. Changes in C18 and >C18 fatty acid levels reflect very long chain fatty acid (VLCFA) elongase activity. Chain length is associated with the degree of unsaturation in VLCFAs[19]. Physical interactions between the VLCFA-elongase complex and desaturase have been reported in yeast[20]. Functional crosstalk between the elongase complex and desaturase has been reported in myoblasts[21]. As expected, the acyl chain length and the degree of unsaturation were correlated. Aged muscle increased in C18 acyl chain levels, but >C18 acyl chains decreased (Fig. 1i). Taken together, aged skeletal muscle exhibits a unique lipidomic signature of acyl chain saturation. These results prompted us to investigate whether FABP3 upregulation in aged skeletal muscle might be involved in such lipid remodeling.

**FABP3-overexpressing young muscle exhibits a lipid composition similar to aged muscle.** To study the impact of FABP3 on intramuscular lipid composition, we transfected plasmid constructs encoding HA-cherry FABP3 or HA-cherry control into young TA muscles using an electroporation gene delivery system (Supplementary Fig. 2a). Using lipidomic analysis, we found that FABP3 overexpression increased 53 lipid species and decreased 32 lipid species (Fig. 2a). Lipid species with significant changes comprised nearly half of the total identified lipid species, indicating that FABP3 plays a major role in muscle lipidome remodeling. We next analyzed lipid class distribution and alteration (Supplementary Table 1). Compared to young muscle controls, FABP3 overexpression increased total SM and LPC contents by 29 and 42%, respectively (Fig. 2b, Supplementary Fig. 2b and Supplementary Table 1). Interestingly, this pattern was similar in aged muscle (Fig. 1c).

We then analyzed phospholipid acyl chain composition. Interestingly, PCs containing polyunsaturated acyl chains such as 16:0/22:6, 18:0/22:6, 18:2/22:6, and 18:1/22:6 decreased in FABP3-overexpressing muscle compared to control young muscle. In contrast, PCs containing 18:1/18:2, 18:0/18:2, and 16:0/16:0 acyl chains increased (Fig. 2c and Supplementary Fig. 2c). When the PC species with the same number of double bonds were combined, PCs containing more double bonds decreased in FABP3-overexpressing muscle, whereas PCs with fewer double bonds increased (Supplementary Fig. 2d). Saturated

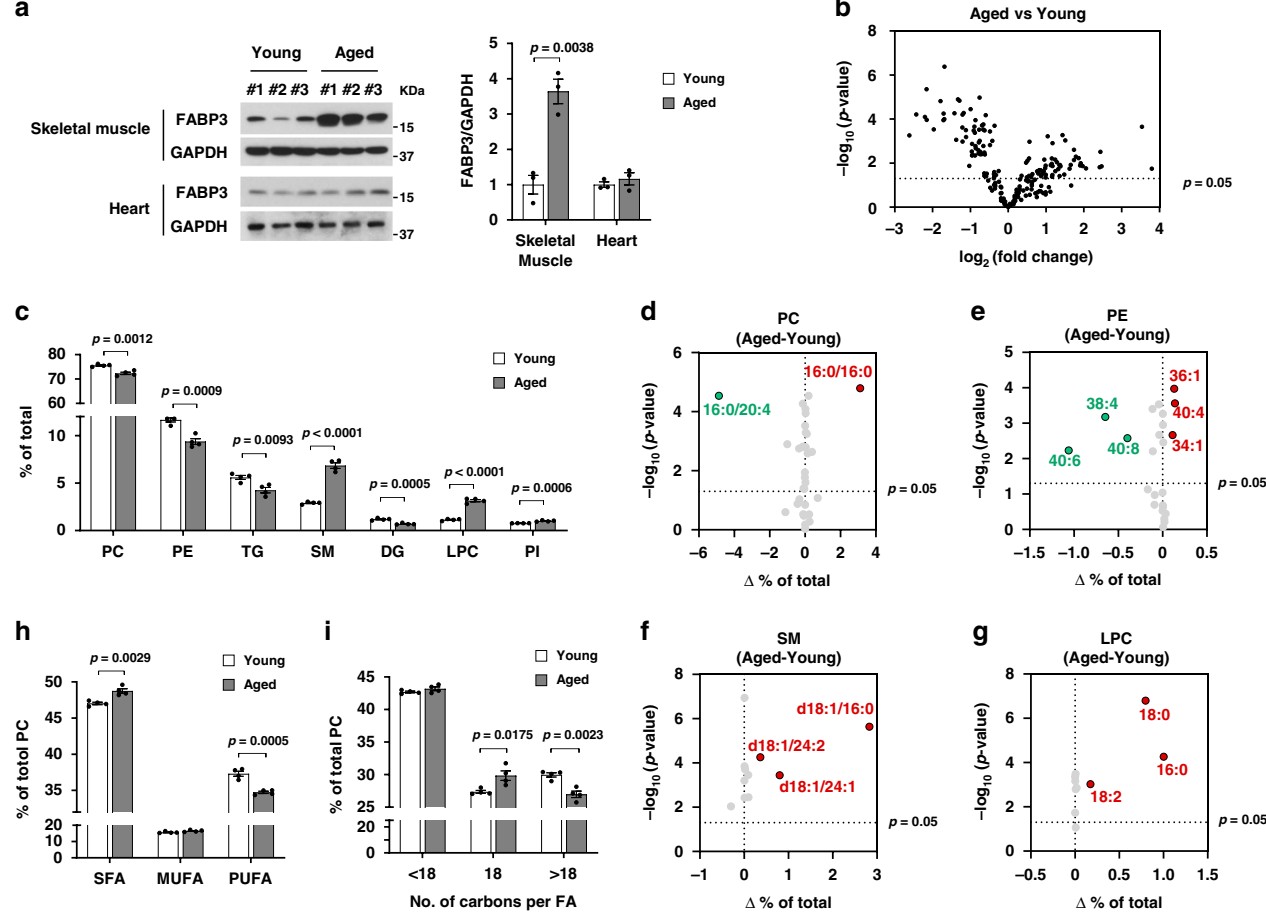

**Fig. 1 Lipidomic signature of aged muscle. a** Immunoblots (top) and quantification (bottom) of the indicated proteins in skeletal muscle and heart isolated from young and aged mice ($n = 3$ mice per group). **b** Volcano plot of lipid species altered in aged vs. young muscle ($n = 4$ mice per group). Lipid species were measured by LC-MS. The $x$-axis indicates the logarithmic (base 2) fold abundance changes of all identified lipid species and the $y$-axis indicates negative logarithmic (base 10) $t$-test $p$-value. The horizontal dotted line reflects the filtering criterion ($p$-value $= 0.05$). **c** Proportion of major lipid classes in young and aged muscles. **d-g** Volcano plots of PC (**d**), PE (**e**), SM (**f**), and LPC (**g**) species altered in aged vs. young muscles. The $x$-axis indicates the percentage changes (aged-young) in lipid abundance and the $y$-axis indicates negative logarithmic (base 10) $t$-test $p$-values. The horizontal dotted line reflects the filtering criterion ($p$-value $= 0.05$). Red and green indicate highly increased and decreased lipid species, respectively. **h** Proportion of saturated (SFA), monounsaturated (MUFA), or polyunsaturated (PUFA) PC acyl chains in young and aged muscles. **i** Proportion of <C18, C18, or >C18 PC acyl chains in young and aged muscles. Data are presented as means ± S.E.M. Two-tailed unpaired Student's $t$-test was used. Source data are provided as a Source Data file.

PC acyl chains increased in FABP3-overexpressing muscle, while polyunsaturated acyl chains decreased (Fig. 2g). Polyunsaturated 40:6, 40:7, and 44:10 PEs significantly decreased, but 36:2, 36:4, 36:3, and 34:2 PEs increased in FABP3-overexpressing muscle (Fig. 2d and Supplementary Fig. 2e). When the PE species containing the same number of double bonds were combined, PEs with more double bonds decreased in FABP3-overexpressing muscle, whereas PEs containing fewer double bonds increased, indicating that PE acyl chains were shifted toward saturation in FABP3-overexpressing muscle (Supplementary Fig. 2f). This pattern of phospholipid saturation in FABP3-overexpressing muscle was similar in aged muscle. SM and LPC species increased in FABP3-overexpressing muscle (Fig. 2e, f and Supplementary Fig. 2g, h). Regarding PC acyl chain length, C18 acyl chains increased in FABP3-overexpressing muscle, while >C18 acyl chains decreased, similar to aged muscle (Fig. 2h). Finally, we analyzed the correlation of the up/down ratio of all identified lipid species between aged and FABP3-overexpressing muscles in comparison with young muscles (Fig. 2i). Surprisingly, a strong linear correlation was found in lipid composition between aged muscle and FABP3-overexpressing young muscle. Based

on these lipidomic data, we suggest that FABP3 contributes to lipid alterations during muscle aging.

**FABP3 executes PERK–eIF2α-mediated inhibition of protein translation.** Given that increased membrane lipid saturation induces ER stress and the unfolded protein response[22], we investigated whether ER stress was induced in aged mouse muscle with elevated membrane SFA content. Phosphorylation of an ER stress sensor, PERK (pancreatic ER kinase), increased in aged TA muscle (Fig. 3a). Consistent with evidence that PERK phosphorylates eIF2α, thereby inhibiting protein translation[23], we detected increased eIF2α phosphorylation and ~50% decreased de novo protein synthesis in aged muscle (Fig. 3a). Therefore, we next investigated whether lipid remodeling via FABP3 overexpression could induce ER stress in vivo. Mouse TA muscle overexpressing FABP3 had increased PERK and eIF2α phosphorylation, and ~40% reduced de novo protein synthesis (Fig. 3b). These results suggest that FABP3-induced lipid remodeling might induce an ER stress response and defective protein synthesis in aged skeletal muscle.

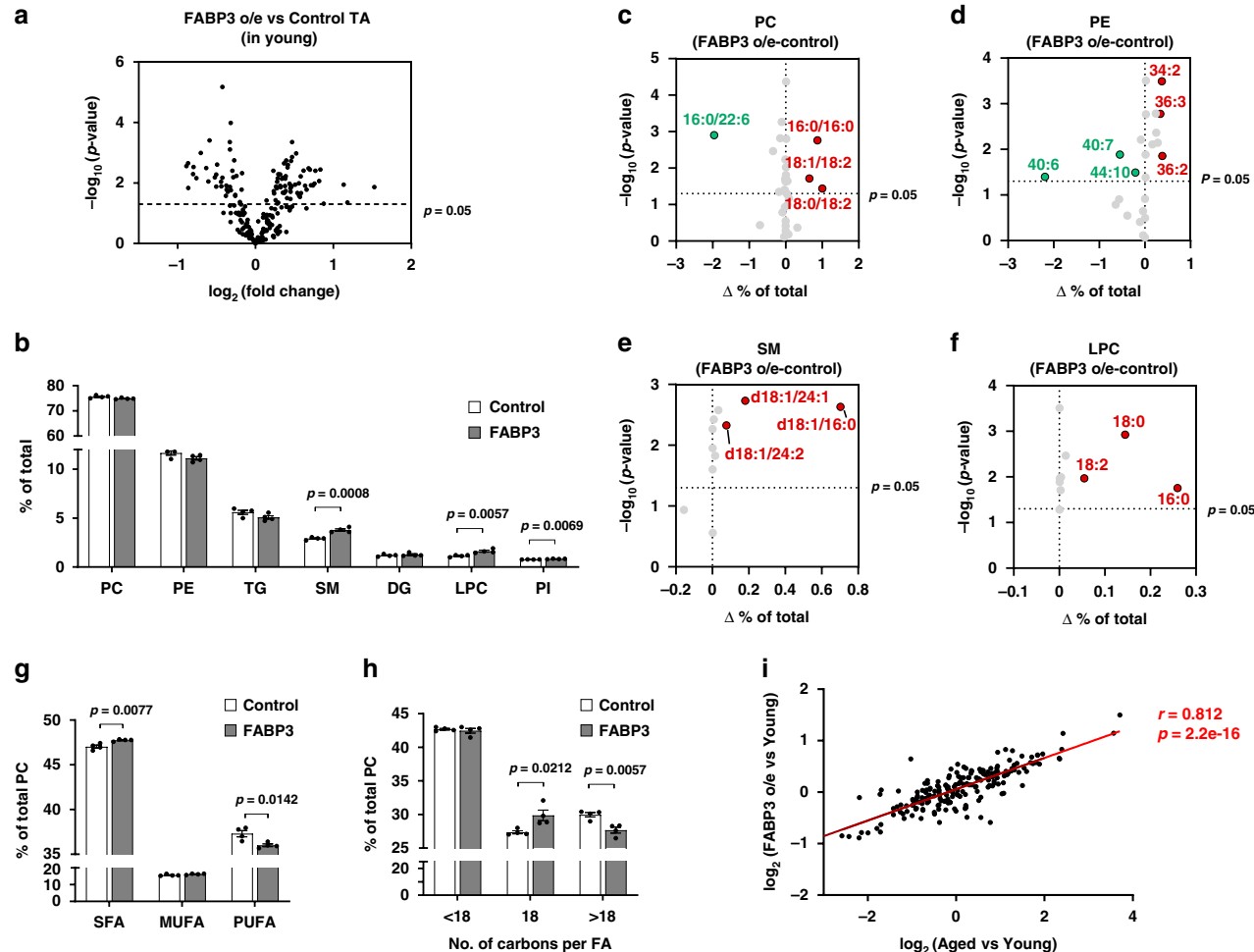

**Fig. 2 FABP3-overexpressing young muscle exhibits lipid composition similar to aged muscle. a** Volcano plot of lipid species altered in FABP3-overexpressing muscle vs. control muscle. Lipid species were measured by LC-MS. The x-axis indicates the logarithmic (base 2) fold abundance changes of all identified lipid species and the y-axis indicates negative logarithmic (base 10) t-test p-value. The horizontal dotted line reflects the filtering criterion (p-value = 0.05). (n = 4 mice per group). **b** Proportion of major lipid classes in FABP3-overexpressing and control muscles. **c–f** Volcano plot of PC (**c**), PE (**d**), SM (**e**), and LPC (**f**) species altered in FABP3-overexpressing vs. control muscles. The x-axis indicates the percentage changes (FABP3 o/e-control) in lipid species and the y-axis indicates negative logarithmic (base 10) t-test p-values. The horizontal dotted line reflects the filtering criterion (p-value = 0.05). Red and green indicate highly increased and decreased lipid species, respectively. **g** Proportion of SFA, MUFA, or PUFA PC acyl chains in FABP3-overexpressing and control muscles. **h** Proportion of <C18, C18, or >C18 PC acyl chains in FABP3-overexpressing and control muscles. Data are presented as means ± S.E.M. Two-tailed unpaired Student's t-test was used. **i** Correlation analysis of the total identified lipid species in the indicated comparative condition. The x-axis indicates logarithmic (base 2) fold concentration changes of all identified lipid species in aged vs. young muscle, and the y-axis indicates logarithmic (base 2) fold changes in FABP3-overexpressing muscle vs. young muscle. Data in **i** were analyzed using Spearman's correlation; correlation coefficient (r) and p-value (p) are in red. Source data are provided as a Source Data file.

To further confirm the role of FABP3 on ER stress signaling and protein synthesis, we established a stable C2C12 cell line that overexpressed FABP3 under the control of Cre recombinase. Acutely overexpressing FABP3 via Cre recombinase-expressing adenovirus increased PERK phosphorylation in differentiated C2C12 myotubes. Further, FABP3 overexpression increased eIF2α phosphorylation and inhibited de novo protein synthesis (Fig. 3c and Supplementary Fig. 3a), which was more severe than in myotubes treated with the well-known ER stress inducer, palmitate. PERK-eIF2α regulates cell death via ATF4 and CHOP activation upon ER stress[24], and inhibits protein synthesis via the 43S preinitiation complex[25]. We excluded a possibility that FABP3 induced cell death via PERK–eIF2α, because FABP3 expression did not induce *Atf4* and *Chop* mRNA expression (Supplementary Fig. 3b). Since we found that PERK inhibition almost totally rescued FABP3-induced inhibition of protein synthesis (Supplementary

Fig. 3c), we postulated that FABP3-driven eIF2α phosphorylation inhibits protein synthesis by directly inhibiting the 43S pre-initiation complex.

Next, we investigated whether alternative ER membrane-associated sensors including inositol requiring transmembrane kinase/endonucleases-1α (IRE-1α) and activating transcription factor 6 (ATF6) could be involved in FABP3-driven ER stress. We found that neither did FABP3 induce phosphorylation of IRE-1α or its downstream effectors, JNK, p65, and SEK, nor did it induce ATF6 cleavage (Supplementary Fig. 3d, f). Moreover, no significant change was observed in the mRNA expression level of unfolded protein response genes, *Xbp1-s, Grp78/Bip Erdj4*, and *Edem*, which are downstream of IRE-1α and ATF6 (Supplementary Fig. 3e). While palmitate induces ER stress via simultaneous activation of the PERK, IRE1α, and ATF6 axis in myotubes[26], our data suggest that FABP3 induces ER stress

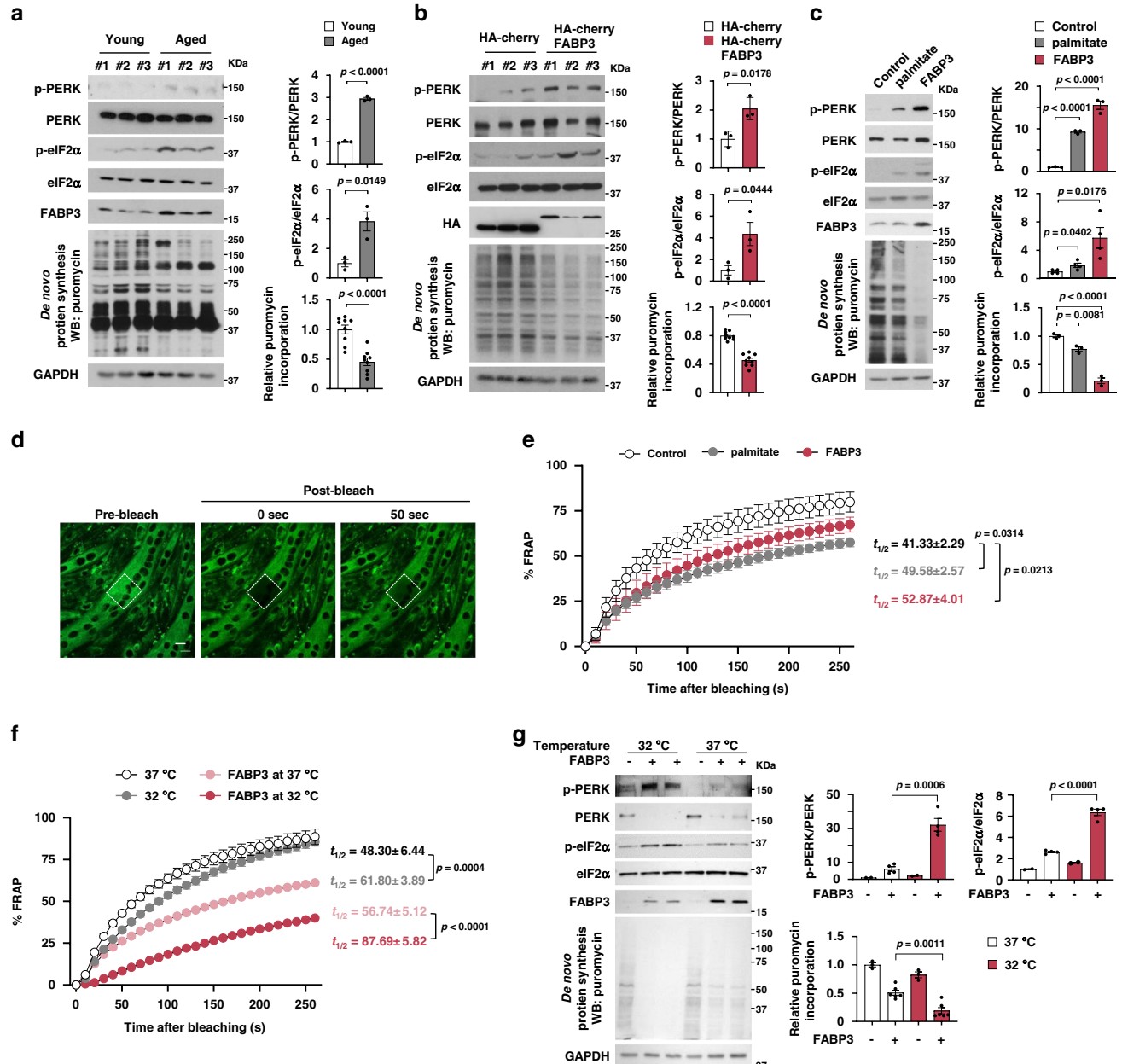

**Fig. 3 FABP3-induced membrane rigidity inhibits de novo protein synthesis by inducing the ER stress response. a–c** Immunoblot analysis (left) and quantification (right) of the PERK and eIF2α phosphorylation and puromycin incorporation in young (6 months) and aged (25 months) TA muscles ($n = 3$ mice per group) (**a**), FABP3-overexpressing and control TA muscles ($n = 3$ mice per group) (**b**), and FABP-overexpressing myotubes ($n = 3$ independent experiments) (**c**). Mice were injected intraperitoneally with puromycin and muscles (**a**, **b**) were harvested 30 min post-injection. TA muscles of young mice (**b**) were transfected with HA-cherry-FABP3 or HA-cherry control plasmid and were harvested 5 days after transfection. Fully differentiated C2C12 myotubes (**c**) expressing Cre-inducible FABP3 constructs were infected with Cre-carrying adenovirus (Ad-Cre) and cultured for 3 days before puromycin treatment. Palmitate was treated for 12 h as a positive control ER stress inducer. **d** Membrane fluidity was analyzed by FRAP. Representative confocal images of the BODIPY 500/510 C1, C12 -labeled myotubes before bleaching (pre-bleach) and at 0 and 50 s. after bleaching (post-bleach). White dotted squares, bleached areas. Scale bar, 20 μm. ($n = 24$ independent experiments). **e** Time course fluorescence gain in palmitate-treated or FABP3-overexpressing C2C12 myotubes. $t_{1/2}$, the half-time for fluorescence recovery; black, control, $n = 9$; gray, palmitate, $n = 7$; red, FABP3, $n = 8$ each myotube. **f**, **g** Temperature effect on membrane fluidity and ER stress. FABP3-overexpressing and control C2C12 myotubes were incubated at 32 or 37 °C. Membrane fluidity (**f**) was measured by FRAP analysis. Note the $t_{1/2}$ values in FRAP analyses. Black, 37 °C, $n = 6$; gray, 32 °C, $n = 8$; red, FABP3 at 32 °C, $n = 6$; light red, FABP3 at 37 °C, $n = 8$ each myotube. **g** Immunoblot analysis (left) and quantification (right) of the indicated proteins and puromycin incorporation. Data are presented as means ± S.E.M. Two-tailed unpaired Student's $t$-test was used. Source data are provided as a Source Data file.

primarily via PERK–eIF2α–43S pre-initiation complex mediated inhibition of protein translation. Meanwhile, either was FABP3-dependent inhibition of de novo protein synthesis associated with AKT–GSK-3β, nor with the mTOR signaling pathway (Supplementary Fig. 3g, h).

**FABP3 overexpression decreased membrane fluidity in myotubes.** Since lipid composition affects membrane fluidity[27], we investigated whether FABP3-induced lipid remodeling could affect muscle cell membrane fluidity. To this end, we measured the membrane lipid diffusion rate by FRAP analysis of 4,4-

difluoro-5-methyl-4-bora-3a,4a-diaza-s-indacene-3-dodecanoic acid (BODIPY 500/510 C1, C12) in myotubes (Fig. 3d). Consistent with a previous report[28], palmitate-treated, BODIPY 500/510 C1, C12-labeled myotubes decreased in fluorescence gain by ~36% at 50 s after bleaching (Fig. 3e). Surprisingly, FABP3-overexpressing myotube membranes had diffusion rates similar to palmitate-treated myotubes. Half-maximal fluorescence gain time in control, palmitate-treated, and FABP3-overexpressing myotubes were 41.33, 49.58, and 52.87 s, respectively (Fig. 3e). These results indicate that FABP3 decreases membrane fluidity. We then sought to determine whether the FABP3-altered membrane fluidity influences ER stress. High temperatures cause membrane fluidization, while low temperatures decrease membrane fluidity[29]. Low temperature (32 °C) markedly aggravated membrane fluidity in FABP3-overexpressing myotubes (Fig. 3f), along with elevating PERK and eIF2α phosphorylation, indicating a more severe ER stress response, and enhanced inhibition of protein synthesis (Fig. 3g). These results suggest that FABP3 induces ER stress by modulating membrane fluidity in aged muscle.

**FABP3 overexpression deteriorates muscle mass and force**. We next investigated the physiological significance of FABP3 levels in skeletal muscle. Muscle recovery after immobilization in aged mice is incomplete or delayed compared to young mice[30]. We, therefore, assessed muscle atrophy after 5 days of immobilization and muscle recovery after 5 days of remobilization in FABP3-overexpressing TA muscle (Fig. 4a). Although the atrophic morphology between FABP3-overexpressing TA muscle and control TA muscle was similar, control young muscle regained muscle mass during the remobilization, while FABP3-overexpressing muscle did not (Fig. 4b). Moreover, immunohistochemical analysis revealed that FABP3-overexpressing muscle displayed significantly smaller muscle fibers than control muscle after remobilization, similar to aged muscle (Fig. 4c, d); meanwhile, western blot analysis revealed that FABP3 maintained high levels of PERK and eIF2α phosphorylation and inhibited protein synthesis during remobilization (Fig. 4e). However, transient FABP3 overexpression per se did not induce a significant difference in muscle mass before immobilization (Supplementary Fig. 4). FABP3 overexpression also did not induce expression of atrophy-related ubiquitin ligases, such as *Atrogin-1* and *MuRF1*, nor impair autophagy by Atg5, 7, 12, 16L, and SQSTM1 expression (Supplementary Fig. 3i–k). Next, to evaluate the effect of FABP3 on skeletal muscle function in intact muscles independent of other parameters, we compared ex vivo contractile properties, force, and fatigability, of FABP3-overexpressing and control young TA muscles. The maximum twitch forces were not different between FABP3-overexpressing and control TA muscles (Fig. 4f). However, at increasing stimulation frequencies in the tetanic force test (10–200 Hz, 100 V), the tetanic force was lower in the FABP3-overexpressing muscle than in the control muscle (Fig. 4g, h), similar to the aged muscle[31]. When subjected to fatigue-inducing repetitive stimulations at 1 Hz, 100 V for 10 min, FABP3-overexpressing muscles were more fatigue-sensitive than control muscles (Fig. 4i). These results together suggest that FABP3 upregulation during aging reduces both muscle mass and force.

**FABP3 inhibition in aged muscle resulted in young-like lipid composition**. To confirm the relevance of FABP3 in age-associated lipid remodeling, FABP3 was knocked down in aged TA muscle using shRNA against FABP3 (Supplementary Fig. 5a). Lipidomic analysis of FABP3 knockdown muscle revealed 64% of total lipid species identified in aged muscle were altered, with 21 lipid species increased and 79 decreased (Fig. 5a). FABP3 knockdown muscle had prominent changes in the lipid classes (Supplementary Table 1). SM and LPC levels decreased by 54 and

53%, respectively (Fig. 5b and Supplementary Fig. 5b, Supplementary Table 1). These results show an inverse correlation with those observed in both FABP3-overexpressing muscle and aged muscle (Figs. 1c, 2b). We next analyzed phospholipid acyl chain composition. Notably, PCs containing polyunsaturated acyl chain such as 16:0/22:6, 18:0/22:6, and 18:2/22:6 increased in FABP3-knockdown muscles (Fig. 5c and Supplementary Fig. 5c), while PCs containing 16:0/16:0, 18:1/18:2, and 18:0/18:2 acyl chains decreased. When PC species containing the same number of double bonds were combined, PCs with more double bonds increased in FABP3-knockdown muscle, whereas PCs containing fewer double bonds decreased (Supplementary Fig. 5d). Polyunsaturated PC acyl chains increased in FABP3-knockdown muscle, while saturated acyl chains decreased (Fig. 5g). Consistently, PEs with more acyl chain double bonds increased, but PEs with fewer double bonds decreased in FABP3-knockdown muscles (Supplementary Fig. 5f). For instance, 40:6, 38:6, and 40:8 PEs increased by 60, 79, and 59%, respectively, while 36:2, 36:3, 34:2, and 34:1 PEs decreased by 35, 50, 32, and 61%, respectively (Fig. 5d and Supplementary Fig. 5e). This phospholipid desaturation pattern in FABP3-knockdown muscle contrasted with FABP3-overexpressing muscle and aged muscle. SM and LPC species decreased simultaneously in FABP3-knockdown muscle (Fig. 5e, f and Supplementary Fig. 5g, h). Additionally, FABP3 knockdown reduced PC C18 acyl chain content but increased >C18 acyl chain content (Fig. 5h). These results were also opposite to those in both FABP3-overexpressing muscle and aged muscle (Figs. 1, 2). The up/downregulation of all identified lipid species displayed a strong correlation between FABP3-knockdown muscles and young muscles in comparison with aged muscles (Fig. 5i). These results together indicate that FABP3 inhibition could recapitulate a young muscle-like lipid composition in aged muscles. Further, principal component analysis (PCA) of total identified lipid species revealed unique lipidomic signatures in young versus aged muscles and in FABP3-overexpressing or FABP3-knockdown muscles. FABP3 overexpression in the young muscle led to clustering toward aged muscles, while FABP knockdown in aged muscles led to clustering toward young muscles (Supplementary Fig. 5i), suggesting that FABP3 drives age-dependent lipid remodeling.

**FABP3 inhibition ameliorated membrane fluidity and alleviated ER stress**. To investigate whether inhibition of FABP3 could protect against ER stress, we examined FABP3 knockdown in aged mouse muscle with ER stress. FABP3 knockdown decreased PERK and eIF2α phosphorylation and improved protein synthesis (Fig. 6a). Consistently in C2C12 myotubes, FABP3 knockdown reduced palmitate-induced PERK–eIF2α phosphorylation and improved protein synthesis (Fig. 6b). To investigate whether FABP3 inhibition could re-establish membrane fluidity that was lost after palmitate treatment, we performed FRAP analysis in FABP3-knockdown C2C12 myotubes. The FRAP signal was significantly increased in FABP3 knockdown myotubes in the presence of palmitate compared to control cells (light blue vs. deep blue symbol in Fig. 6c). Taken together with FABP3 overexpression data showing decreased FRAP signal (Fig. 3), we propose that increased FABP3 in aged muscle may reduce membrane fluidity and the ER stress response.

To test whether lipid remodeling is an underlying mechanism for FABP3-induced membrane rigidity and ER stress, we evaluated the effect of polyunsaturated fatty acids (PUFAs) in FABP3-overexpressing C2C12 myotubes. Addition of docosahexaenoic acid (22:6, DHA) into the culture media markedly enhanced the FRAP signal that was lost both in FABP3-overexpressing (light red vs. deep red symbol) and palmitate-

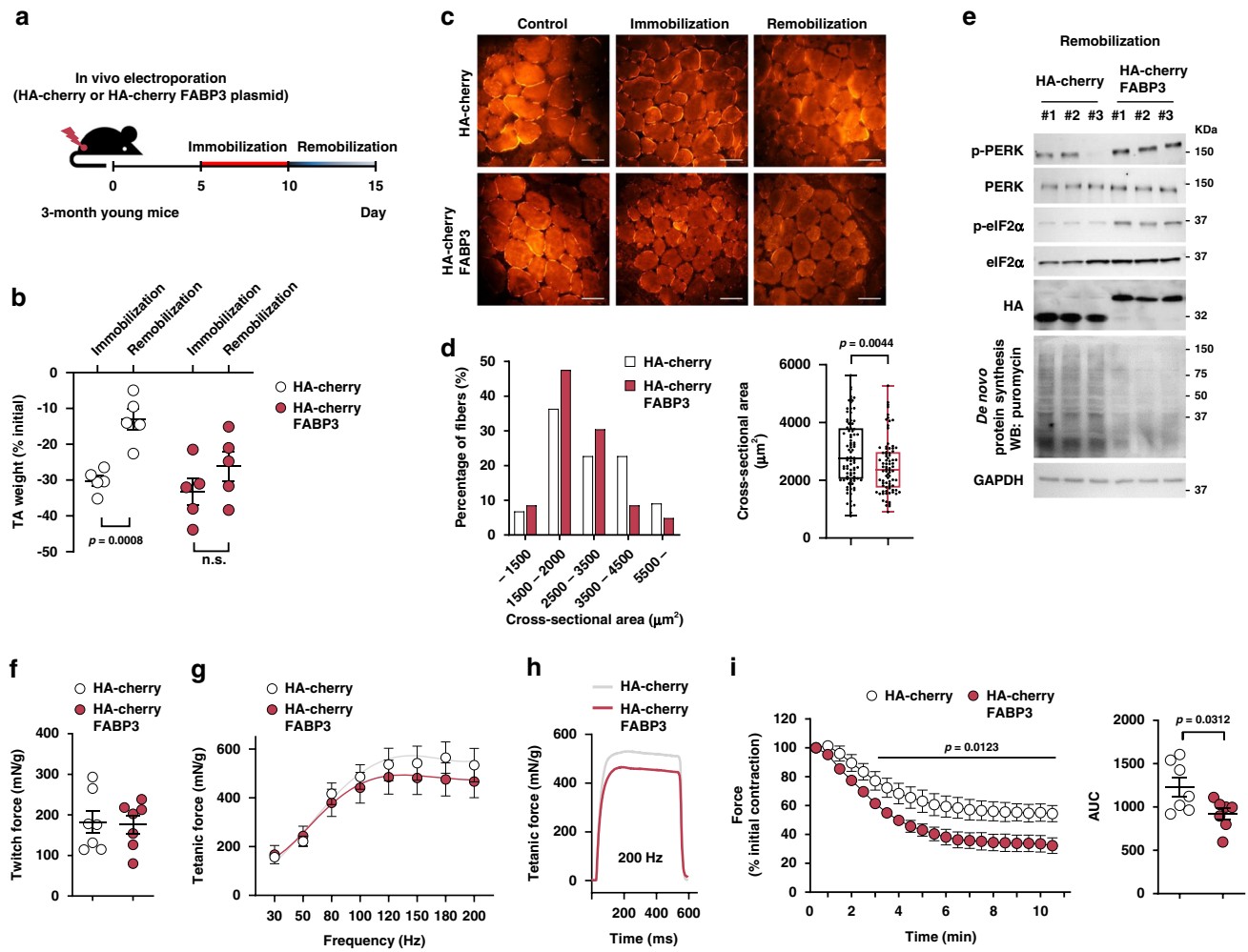

**Fig. 4 FABP3-overexpressing young muscle exhibits impaired muscle recovery after immobilization and early fatigue. a** Scheme of the experimental procedures. TA muscles of young mice were transfected with HA-cherry-FABP3 or HA-cherry control plasmid, immobilized with a surgical staple 5 days after transfection, and allowed to recover (remobilization, $n = 5$) after 5-day immobilization ($n = 5$). **b** TA muscle mass was measured 5 days after immobilization and 5 days after remobilization. Changes in muscle mass were expressed as percentage of control. **c** Representative images of the transfected myofibers (red). ($n = 5$ per group). Scale bars, 100 μm. **d** Frequency histograms of the cross-sectional area (left) of HA-cherry-FABP3- or HA-cherry control plasmid-transfected muscle fibers 5 days after remobilization. Box plot representing the mean cross-sectional areas (right) of transfected muscle fibers. Box represent the 25th–75th percentiles of the data; whiskers show the min and max range of the data; horizontal lines indicate the median value. **e** Immunoblot analysis of PERK and eIF2α phosphorylation, and puromycin incorporation in FABP3-overexpressing and control TA muscles after remobilization. Five days after remobilization, mice were injected intraperitoneally with puromycin. TA muscles were harvested 30 min post-injection ($n = 3$ mice per group). **f–i** Muscle forces were measured in intact HA-cherry FABP3 or HA-cherry control plasmid-transfected TA muscles mounted on a force transducer. The maximum twitch force (**f**) at supramaximal voltage, 100 V for 1 ms. Frequency dependence (at 30–200 Hz, 100 V, 500 ms) of average tetanic force (**g**). Tetanic force traces (**h**) at 200 Hz for 500 ms ($n = 7$ mice per group). Fatigue index (**i**) was measured at 1 Hz and 100 V for 10 min. Generated force was recorded and expressed as a percentage of the initial force (left). Insert represents area under curve (AUC) (right) ($n = 7$ mice per group). Data are presented as means ± S.E.M. Two way ANOVA withBonferroni's post hoc test was used (left in **i**) and two-tailed unpaired Student's *t*-test was used (**b**, **d**, right in **i**). Source data are provided as a Source Data file.

treated myotubes (light blue vs. deep blue symbol) (Fig. 7a). These results suggest that PUFAs restore membrane fluidity in such myotubes. DHA supplementation also significantly suppressed FABP3-induced PERK and eIF2α phosphorylation, while improving protein synthesis (Fig. 7b), resulting in improved myotube recovery that was defective in FABP3-overexpressing myotubes (Fig. 7c). This result is consistent with previous reports showing DHA inhibits palmitate-induced ER stress in C2C12 myotubes[26]. We suggest that FABP3-induced lipid remodeling may be an underlying mechanism in age-related membrane rigidity and ER stress, which is partially rescued by increased PUFA content.

**FABP3 knockdown ameliorated age-associated impairment of muscle recovery.** We investigated whether FABP3 knockdown ameliorates impaired TA muscle recovery in aged mice (Fig. 8a). FABP3-knockdown tended to slow the muscle loss during immobilization, and significantly increased muscle mass during remobilization. (Fig. 8b). Immunohistochemical analysis revealed significantly larger muscle fibers in FABP3-knockdown aged muscle than in control aged muscle after remobilization (Fig. 8c, d). FABP3-knockdown maintained low levels of PERK and eIF2α phosphorylation and improved protein synthesis during remobilization (Fig. 8e). Next, to evaluate the effect of FABP3 inhibition on aged skeletal muscle function, we compared the ex vivo muscle contractility of FABP3-knockdown and aged muscle.

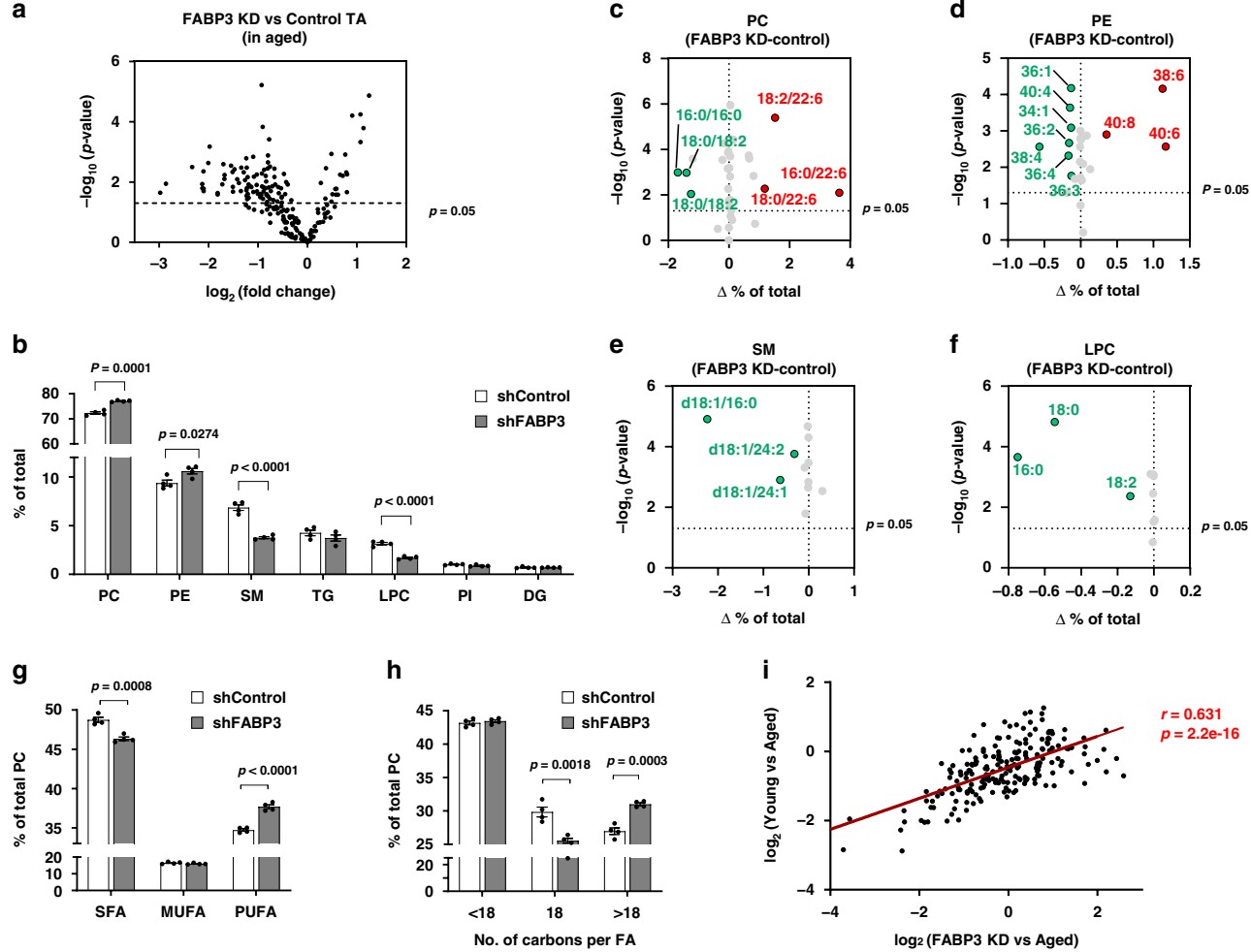

**Fig. 5 FABP3 inhibition in aged muscle results in young-like lipid composition. a** Volcano plot of lipid species altered in FABP3-knockdown aged vs. control muscle. Lipid species were measured by LC-MS. The x-axis indicates logarithmic (base 2) fold abundance changes of all identified lipid species and the y-axis indicates negative logarithmic (base 10) t-test p-value. The horizontal dotted line reflects the filtering criterion (p-value = 0.05). (n = 4 mice per group) **b** Proportion of major lipid classes in FABP3-knockdown and control aged muscles. **c–f** Volcano plot of PC (**c**), PE (**d**), SM (**e**), LPC (**f**) species altered in FABP3-knockdown vs. control aged muscles. The x-axis indicates the percentage changes (FABP3KD-control-aged muscle) of lipid species and the y-axis indicates negative logarithmic (base 10) t-test p-values. The horizontal dotted line reflects the filtering criterion (p-value = 0.05). Red and green indicate highly increased and decreased lipid species, respectively. **g** Proportion of SFA, MUFA, or PUFA PC acyl chains in FABP3-knockdown and control-aged muscles. **h** Proportion of <C18, C18, or >C18 PC acyl chains in FABP3-knockdown and control-aged muscles. The data are presented as means ± S.E.M. Two-tailed unpaired Student's t-test was used. **i** Correlation analysis of the total identified lipid species in the indicated comparative condition. The x-axis indicates logarithmic (base 2) fold concentration changes of all identified lipid species in FABP3-knockdown aged vs. aged muscle and the y-axis indicates logarithmic (base 2) fold changes in aged vs. young muscle. Data in **i** were analyzed using Spearman's correlation; correlation coefficient (r) and p-value (p) are in red. Source data are provided as a Source Data file.

While the maximum twitch force remained unchanged (Fig. 8f), the tetanic force at high frequencies (150–200 Hz) was slightly higher in FABP3-knockdown muscle than in control aged muscle (Fig. 8g, h). While aged muscles displayed a gradual reduction in tetanic forces over 120 Hz, FABP3-knockdown muscle retained tetanic force up to 200 Hz (Fig. 8g). When subjected to fatigue-inducing repetitive stimulations, FABP3-knockdown muscle was more fatigue-resistant than aged muscles (Fig. 8i). Together, these results indicate that FABP knockdown ameliorates age-related decline in muscle mass and strength, thus serving as a potentially valuable therapeutic target.

## Discussion
In this study, we demonstrated that FABP3 upregulation in aged muscle could be an intrinsic cue contributing to the age-associated loss of muscle mass and strength via regulation of membrane lipid composition. FABP3-overexpressing young muscle exhibited lipid composition similar to aged muscle and had delayed recovery after muscle immobilization, with defective protein synthesis induced by ER stress. Interestingly, FABP3 inhibition improved intrinsic ER-stress-induced muscle dysfunction in aged muscle, with a noticeable restoration of young muscle-like lipid composition. We revealed that proper FABP3 regulation is critical for maintenance of membrane fluidity. Collectively, our results suggest that FABP3 is a key target that modulates age-dependent lipid remodeling and subsequent muscle homeostasis.

Lipidomic analyses of mouse hindlimb muscles revealed that PUFA/SFA levels were decreased among the membrane phospholipids in both aged muscles and FABP3-overexpressing young muscles in comparison with the control young muscle, yet higher in the FABP3-knockdown aged muscle (Figs. 1, 2, and 5). We excluded the possibility that FABP3 alters membrane lipid composition

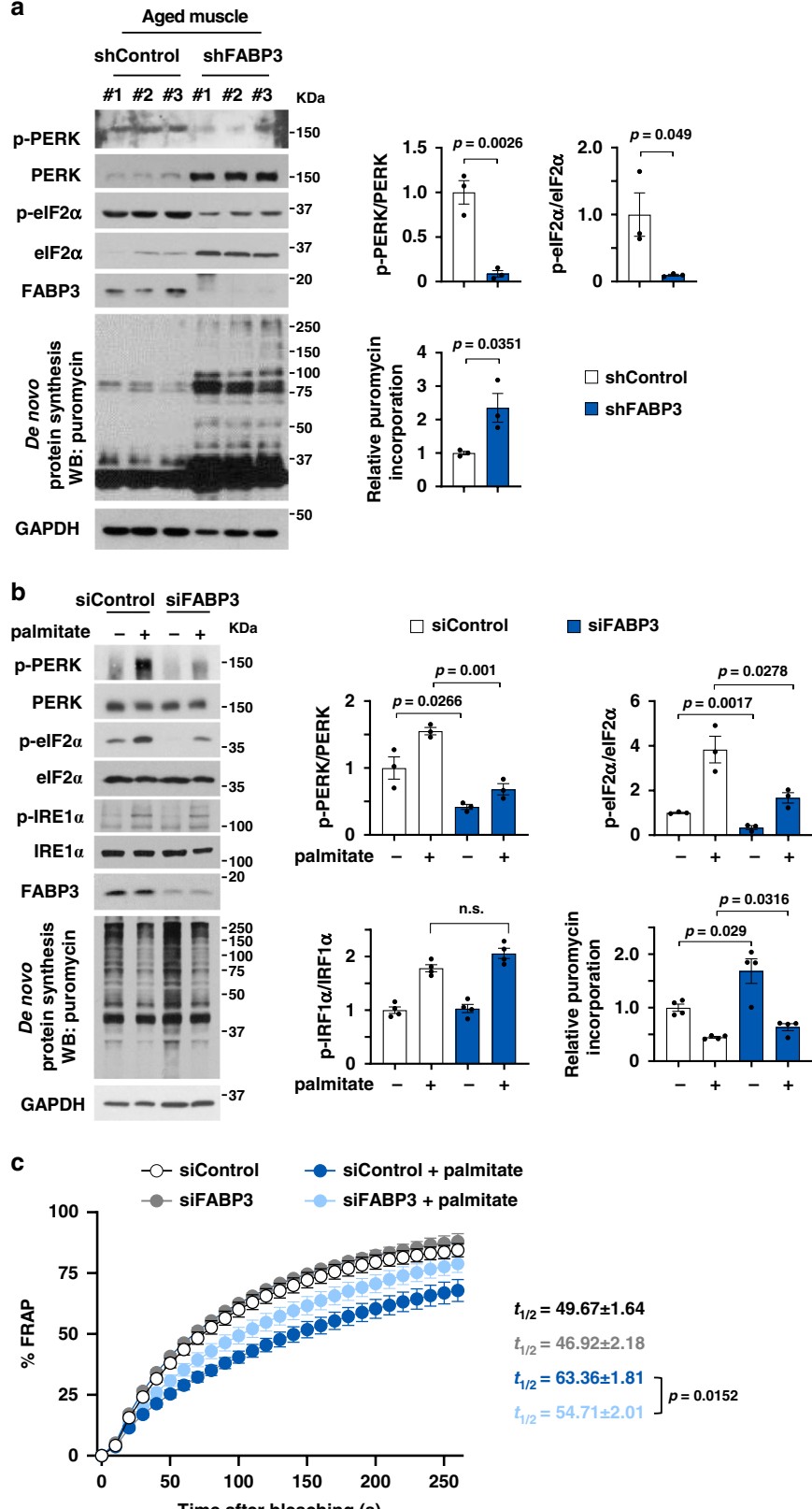

**Fig. 6 FABP3 inhibition alleviated ER stress. a**, **b** Immunoblot analysis (left) and quantification (right) of the indicated proteins and puromycin incorporation in FABP3-knockdown aged and control muscles (**a**) and FABP3-knockdown myotubes (**b**). GAPDH was used as a loading control. TA muscles of aged mice (**a**) were infected with Ad-shFABP3 virus or Ad-shControl. Mice were injected intraperitoneally with puromycin (0.04 mmol/kg) 5 days after infection (n = 3 mice per group). C2C12 myotubes (**b**, **c**) were transfected with siFABP3 or siControl and treated with palmitate (500 μM) or vehicle for 12 h. Myotubes were then incubated with puromycin (1 μM) for 30 min (n = 3 independent experiments) (**b**). **c** Time course fluorescence recovery. Note the $t_{1/2}$ values in FRAP analyses. Black, siControl, n = 6; gray, siFABP3, n = 6; deep blue, siControl + palmitate, n = 6; light blue, siFABP3+palmitate, n = 7 each myotube. Data are presented as means ± S.E.M. Two-tailed unpaired Student's t-test was used. Source data are provided as a Source Data file.

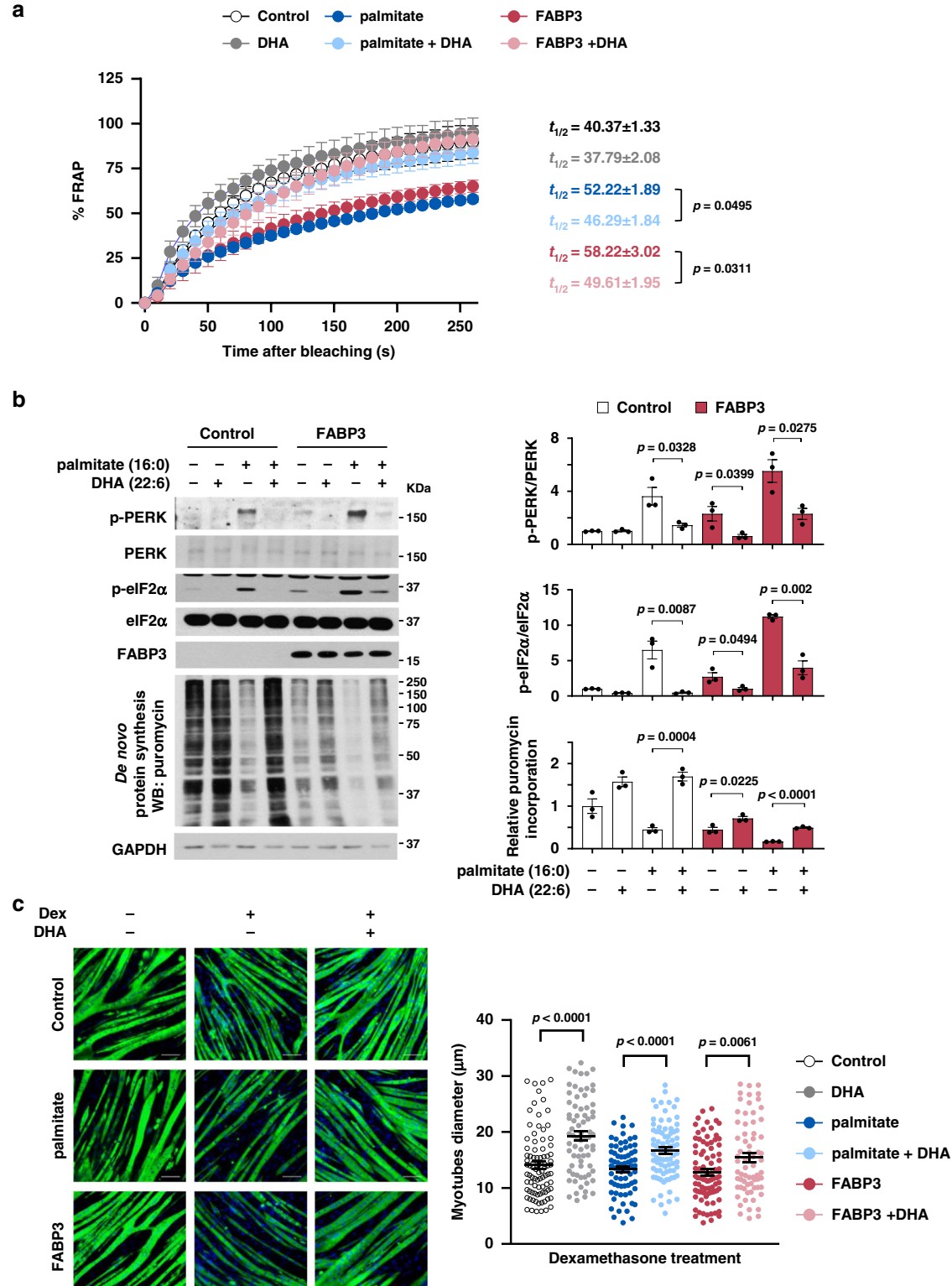

by regulating mRNA expression of fatty acid synthesizing enzymes (Supplementary Fig 6a, b). Therefore, we believe that the composition is altered owing to the higher affinity of FABP3 for SFAs compared to PUFAs[15], leading to subsequent delivery of SFAs to the membrane[32]. In humans, increasing evidence indicates the beneficial effect of fish oil-derived $n − 3$ PUFA supplements on muscle mass and function[33], probably by exerting anti-inflammatory effects, through mTOR activation, and through an

increase in insulin sensitivity. Anti-inflammatory effects would potentially result from a reduction in pro-inflammatory cytokine levels, such as CRP, IL-6, IL-1β, and TNFα. Activated insulin and mTOR signaling pathways are probably involved in the increase in muscle protein synthesis; however, the underlying mechanism is yet unclear[34]. We observed that DHA supplementation in FABP3-overexpressing myotubes alleviated PERK-mediated ER stress and restored protein synthesis (Fig. 7). Taken together, we suggest that

**Fig. 7 PUFA supplementation alleviates FABP3-induced ER stress.** Effects of DHA on fluidity (**a**), ER stress signal (**b**), and morphology (**c**). FABP3 expression was induced with Cre recombinase-carrying adenovirus (Ad-Cre) infected into fully differentiated C2C12 myotubes harboring Cre-inducible FABP3 constructs. FABP3-overexpressing myotubes or palmitate (500 μM)-treated myotubes were treated with DHA (100 μM) or vehicle. **a** Membrane fluidity was measured by FRAP analysis. Note the $t_{1/2}$ values in FRAP analyses. Black, control, $n = 6$; gray, DHA, $n = 6$; deep blue, palmitate, $n = 5$; light blue, palmitate+DHA, $n = 9$; red, FABP3, $n = 5$; light red, FABP3+DHA, $n = 9$ each myotube. **b** Immunoblot analysis (left) and quantification (right) of the indicated proteins and puromycin incorporation. Myotubes were incubated with puromycin (1 μM) for 30 min ($n = 3$ independent experiments). **c** the effect of DHA supplementation on FABP3-induced loss of atrophy recovery. FABP3-overexpressing or palmitate-treated myotubes were treated with dexamethasone (10 μM). Representative images (left) of MyHC (green) and DAPI (blue) staining in myotubes following DHA supplementation (100 μM) for 24 h ($n = 3$ independent experiments). Scale bar, 100 μm. Quantification (right) of myotubes diameter treated with dexamethasone. Data are presented as means ± S.E.M. Two-tailed unpaired Student's $t$-test was used. Source data are provided as a Source Data file.

age-related FABP3 upregulation could disrupt muscle homeostasis by decreasing PUFA/SFA content of muscle membrane.

We investigated how phospholipid saturation induces ER stress and subsequent dysfunction in aged muscle. The PUFA/SFA ratio is a key factor determining membrane fluidity[27]. Membrane lipid saturation induces IRE1α and PERK activation and subsequent ER stress responses[22]. Combining these reports, we hypothesized that FABP3-mediated membrane lipid saturation in aged muscle might induce membrane rigidity, which in turn leads to ER stress. In this study, we demonstrate that FABP3 overexpression decreases membrane fluidity, while FABP3 knockdown increases membrane fluidity (Figs. 3e and 6c), which suggests a membrane fluidity-related mechanism in FABP3-induced ER stress in aged muscle.

Another noticeable lipid change was that SM and LPC content increased in aged and FABP3-overexpressing young muscle but decreased in FABP3-knockdown aged muscle. Sphingolipids promote muscle wasting and contractile dysfunction through induction of apoptosis and the inhibition of progenitor cell proliferation, insulin receptor signaling, glucose and amino acid uptake, and protein synthesis[35,36]. LPC inhibits myoblast fusion and decreases muscle fiber growth by reducing membrane fluidity[21,37]. Considering that both SM and LPC contain mostly saturated or monounsaturated acyl chains (see ref. [18] and Figs. 1f, g, 2e, f, and 5e, f), it is likely that increased SM and LPC content in aged or FABP3-overexpressing muscle (Figs. 1c, 2b) contributes to membrane saturation and subsequent reduction in membrane fluidity.

Here, we showed the mechanical link between membrane fluidity and ER stress by temperature shift assay and showed that activation of PERK, an ER transmembrane protein, is necessary for this link. However, we still do not know the exact mechanism of PERK activation. The lipid composition reportedly influences membrane proteins (e.g., channels or receptors) by modifying their conformation, distribution, or oligomerization[38–40]. We hypothesize that a physical membrane state induced by lipid remodeling causes clustering of PERK and subsequent activation by autophosphorylation, which requires further investigations. Other membranous organelle(s) can be an alternative target of lipid remodeling. To better understand lipid-mediated muscle aging, it is necessary to identify an unknown target and its molecular and physiological functions.

In this study, while we found that increased FABP3 expression in aged muscle is critical for muscle recovery, the molecular mechanism of its increase with aging still remains unclear. We found that SIRT1 negatively regulates FABP3 expression. SIRT1 inhibition by gene silencing, or treatment with chemical inhibitors, such as EX-527 and sirtinol, increased both protein and mRNA levels of FABP3 (unpublished data). It was reported that SIRT1 activity depends on NAD+ levels, which decline with age in skeletal muscle[41]. Therefore, we speculate that decreasing activity of SIRT1 with aging gradually increases FABP3 expression in the skeletal muscle. However, further studies are required to better understand the regulatory mechanism.

Based on our present study showing FABP3-induced alterations in membrane lipid composition and membrane fluidity, we propose a role of FABP3 as a lipid chaperone for the regulation of skeletal muscle aging. Modified membrane lipid composition might change membrane properties, promote ER stress, and decrease protein synthesis, eventually leading to defective muscle mass and force (Fig. 8j). Collectively, we suggest that FABP3 would be a valuable therapeutic target for intervention of sarcopenia.

## Methods

**Animal models**. All animal experiments were performed according to protocols approved by the Animal Care and Use Committee of the Korea Research Institute of Bioscience and Biotechnology (KRIBB). Young naive (3-month-old) and aged naive (22–24-month-old) C57BL/6 mice were purchased from the Laboratory Animal Resource Center (KRIBB). The mice were fed standard chow (Teklad F6 Rodent Diet 8664, Harlan Teklad, Indianapolis, IN), and housed under controlled temperature at 22–24 °C and a 12 h light/12 h dark cycle with a humidity between 40 and 60%. Plasmid DNA was transfected into mouse TA muscle using an electric pulse mediated gene transfer technique. Young mice were anesthetized with iso-flurane, and 50 μg of plasmid DNA containing either HA-cherry-FABP3 or HA-cherry in 50 μl with saline was injected into TA muscle with a 27-gauge needle. An electric field was then applied to the muscle using an electric pulse generator (Electro Square Parator ECM830; BTX). TA muscle overexpressing FABP3 was collected 5 days after transfection. For FABP3 knockdown, TA muscle of aged mouse was injected with 50 μl Ad-shFABP3 adenovirus ($10^8$ CFU, shADV-258510; Vector Biolabs) or shControl (1782; Vector Biolabs). FABP3-knockdown TA muscles of aged mice were collected 5 days after infection. The contralateral TA was used as an intrasubject control. To induce atrophy and recovery, the hindlimb was immobilized for 5 days by stapling the foot, exploiting normal dorsotibial flexion using an Autosuture Royal 35 W skin stapler[42]. One tine was inserted close to the toe on the plantar portion of the foot, while the other was inserted in the distal portion of the gastrocnemius. The hindlimb was remobilized for 5 days by removing the tine. Mice were sacrificed, and isolated muscle tissues were used for further analysis.

**Cell culture**. C2C12 mouse muscle cell line (ATCC, #CRL-1772, female) was maintained in 5% $CO_2$ at 37 °C and grown in DMEM containing 10% fetal bovine serum, penicillin (100 U/ml), and streptomycin (100 μg/ml). C2C12 cells were differentiated into myotubes by changing to medium containing 2% horse serum, penicillin (100 U/ml), and streptomycin (100 μg/ml). To overexpress FABP3 in mature myotubes, Cre-inducible stable cell lines (pCCALL2-LacZ/FABP3) were generated and cultured. The stable cell line was cultured in DMEM containing 10% FBS, penicillin (100 U/ml), streptomycin (100 μg/ml), and 500 μg/ml G418 at 37 °C and 5% $CO_2$. The stable cells were differentiated into myotubes for 3 days. Fully differentiated C2C12 myotubes were infected with adenovirus expressing Cre recombinase. FABP3 expression was monitored from 12 to 72 h postinfection. To knockdown FABP3 in mature myotubes, C2C12 cells were differentiated into myotubes by changing to differentiation medium containing 2% horse serum, penicillin (100 U/ml), and streptomycin (100 μg/ml). After 3 days in differentiation media, 50 pmol siRNA against FABP3 or siControl was transfected into myotubes using RNAiMAX (Invitrogen). For palmitate treatment, sodium palmitate (Sigma-Aldrich) was dissolved in 100% ethanol while incubating in a 55 °C water bath, and then mixed with 2% fatty acid-free BSA in DMEM. C2C12 myotubes were treated with 500 μM palmitate or 100 μM DHA (Sigma-Aldrich). C2C12 myotubes were treated with 10 μM dexamethasone (Sigma-Aldrich). C2C12 myotubes were treated with 1 μM GSK2606414 (Merck Millipore).

**Immunofluorescence**. C2C12 myoblasts were seeded on 6-well plates with cover-slips and differentiated into myotubes. Myotubes were rinsed in PBS, fixed in 4% paraformaldehyde for 15 min, and then washed three times with PBS and permeabilized in 0.1% Triton X-100 in PBS for 15 min. The myotubes blocked with 0.2% BSA for 30 min and then incubated with anti-MYH (sc-376157, Santa Cruz

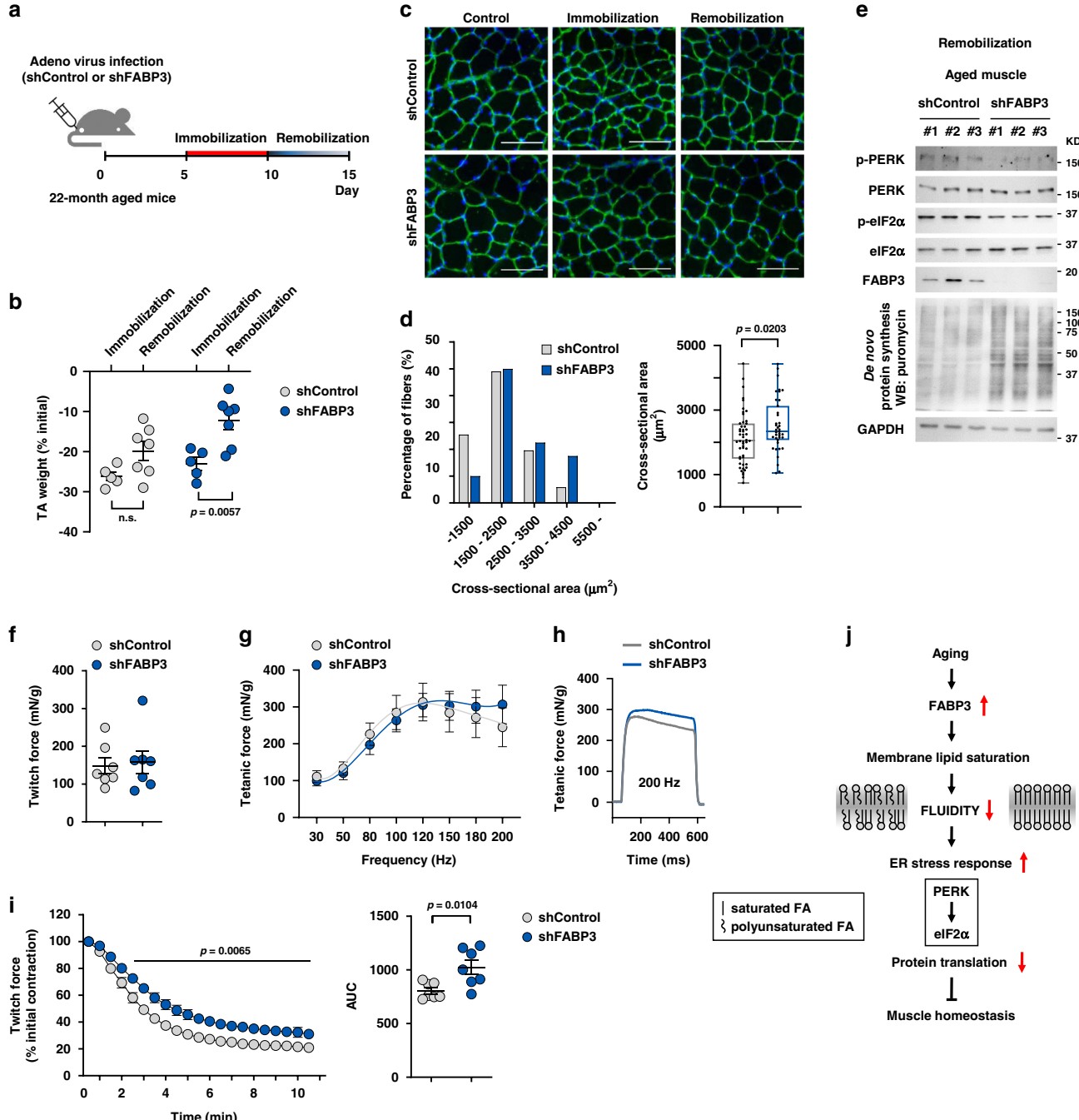

**Fig. 8 FABP3 knockdown ameliorated an impaired muscle recovery in aged mice. a** Scheme of the experimental procedures. TA muscles of aged mice were infected with Ad-shFABP3 or Ad-shControl virus, immobilized with a surgical staple 5 days after infection, and allowed to recover (remobilization) after 5-day immobilization. **b** TA muscle mass was measured at 5 days after immobilization ($n = 5$) and 5 days after remobilization ($n = 7$). Changes in muscle weight for immobilization and remobilization were expressed as the percentage of the contralateral non-immobilized muscle weight. **c** Representative images of laminin (red) and DAPI (blue) staining of myofibers infected with Ad-shFABP3 or Ad-shControl virus Scale bar, 200 μm. **d** Frequency histograms of the cross-sectional area (left) of Ad-shFABP3 or Ad-shControl virus-infected fibers 5 days after remobilization. Box plot representing the mean fiber cross-sectional areas (right). Box represent the 25th–75th percentiles of the data; whiskers show the min and max range of the data; horizontal lines indicate the median value. **e** Immunoblot analysis of PERK and eIF2α phosphorylation, and puromycin incorporation in Ad-shFABP3 or Ad-shControl virus-infected TA muscles after remobilization. TA muscles were harvested 30 min post-interperitoneal puromycin injection. ($n = 3$ mice per group). **f–i** Ad-shFABP3 or Ad-shControl virus-infected TA muscles were mounted on a force transducer. The maximum twitch force (**f**) at supra-maximal voltage, 100 V for 1 ms. Frequency dependence of average tetanic force curves (**g**) at 30–200 Hz, 100 V, 500 ms for each frequency. Tetanic force traces (**h**) upon stimulation at 200 Hz for 500 ms. Data are presented as means. ($n = 7$ mice per group) Fatigue index (**i**) was measured at 1 Hz and 100 V by repeated stimuli for 10 min. Generated force was analyzed as a percentage of the initial contractile force ($n = 7$ mice per group) (left). Insert represents area under curve (AUC) (right). Data are presented as means ± S.E.M. Two way ANOVA with Bonferroni's post hoc test was used (left in **i**) and two-tailed unpaired Student's *t*-test was used (**b**, **d**, right in **i**). **j** Proposed model for age-dependent lipid remodeling by FABP3. Source data are provided as a Source Data file.

Biotechnology) overnight at 4 °C, followed by Alexa Fluor 488 (#A-21121, Thermo-Fisher) for 60 min. Coverslips were mounted with Vectashield with DAPI mounting medium (VECTOR Laboratories). Images were captured using a Nikon Eclipse Ti-U inverted microscope and Nikon DS-Ri2 camera using NIS-Elements software.

**Puromycin incorporation**. To label newly-synthesized proteins with puromycin, C2C12 myotubes were incubated with puromycin (1 μM) for 30 min and harvested. For in vivo labeling, mice were intraperitoneally injected with puromycin (0.04 mmol/kg). TA muscles were harvested after 30 min. Cell or muscle lysates were analyzed using SDS-PAGE followed by immunoblotting. Puromycin-labeled proteins were detected by antipuromycin antibodies (PMY-2A4, DSHB).

**Immunoblot analysis**. Muscle tissues and myotubes were homogenized in lysis buffer containing 20 mM HEPES (pH 7.2), 150 mM NaCl, 0.5% Triton X-100, 0.1 mM Na$_3$VO$_4$, 1 mM NaF, 1 mM 4-(2-aminoethyl)-benzenesulfonyl fluoride hydrochloride (AEBSF), and 5 mg/ml aprotinin (Sigma-Aldrich). The lysates were centrifuged at 15,000×g for 20 min at 4 °C, and the supernatants were subjected to SDS-PAGE followed by immunoblot analysis. Antibodies used are as follows: 4EBP1 (#9452), ATG5 (#12994), ATG7 (#8558), ATG12 (#4180), ATG16L (#8089), beclin1 (#3495), eIF2α (#9722), IRE1α (#3294), mTOR (#2983), p65 (#6956), PERK (#3192), S6K (#9202), SEK (#9152), SQSTM1 (#8025) phospho-4EBP1 (#9459), phospho-AKT S473 (#9271), phospho-GSK-3β (#9327), phosphor-mTOR (#5536), phospho-eIF2α (#9721), and phospho-PERK (#3179), phospho-JNK (#9251), phospho-SEK (#9151), phospho-S6K (#9206) from Cell Signaling Technologies; phospho-IRE1α (ab48187) from Abcam; and HA (sc-805), AKT (sc-1618), ATF6 (sc-22799), MYH (B-5, sc-376157), GSK-3β (sc-7291), and FABP3 (sc-58274) from Santa Cruz Biotechnology; and JNK (51-1570) from BD Bioscience; Laminin (L9393) from Sigma-Aldrich. The anti-GAPDH antibody was developed in our laboratory. Data were collected using Automatic X-ray Film Processor (JP-33, JPI America) or iBright FL1500 (Invitrogen).

**Ex vivo isometric force and fatigue measurements**. Intact TA muscles were dissected from the hindlimb of euthanized mice and mounted vertically between a force transducer (Model FT03, Glass Instruments, USA) in an organ bath with platinum electrodes and continuous perfusion with 95% O$_2$ + 5% CO$_2$-saturated Krebs-Ringer solution (118 mM NaCl, 4.75 mM KCl, 24.8 mM NaHCO$_3$, 1.18 mM KH$_2$PO$_4$, 2.5 mM CaCl$_2$·2H$_2$0, 1.18 mM MgSO$_4$, and 10 mM glucose). Optimal muscle stretch was determined by applying a single twitch at supramaximal voltage (100 V for 1 ms) using a previously described protocol with slight modification[43,44], and set at the length that generated maximal twitch force. After 10 min of equilibration, TA muscles were subjected to different force frequencies (tetani with increasing stimulation frequencies at 30–200 Hz every 500 ms with 2-min recovery intervals). The fatigue properties of TA muscles were assessed through repeated stimulation for 10 min at frequency of 1 Hz and 100 V. All experiments were performed at 25 °C. Data acquisition and analysis were performed using LabChart Pro Software (Version 8; AD instruments, Pty Ltd.). Muscle length, diameter, and wet weight were measured at the end of each experiment.

**Lipidomic analysis**. For lipid extraction, 50 mg muscle tissue and 2.8 mm zirconium oxide beads were placed in a homogenizing tube with 500 μl 70% methanol solution. The sample was homogenized twice for 20 s at a speed of 5000 rpm. Next, 300 μl 70% methanol and 400 μl CHCl$_3$ were added and the mixture was vortexed for 1 min. The mixture was incubated at room temperature for 10 min and centrifuged at 13,000 rpm for 10 min. The lower organic layer was collected and dried under N$_2$ gas. The dried samples were reconstituted into 250 μl isopropanol:acetonitrile:water (2:1:1, v/v/v). Each sample was filtered, and 5 μl was used for lipidomic analysis. Lipids were measured using a triple TOF 5600 mass spectrometry (MS) system (SCIEX, Concord, Canada) coupled with ultra-performance liquid chromatography (UPLC) system (Waters, Milford, USA). An acquity UPLC BEH C18 column (1.7 μm × 2.1 × 100 mm) was used for lipid separation. The mobile phase was 10 mM ammonium acetate in acetonitrile:water (4:6, v/v; solvent A) and 10 mM ammonium acetate in acetonitrile:isopropanol (1:9, v/v; solvent B). Samples were eluted at 0.35 ml/min for 20 min. The eluate was analyzed with electrospray ionization (ESI) in positive and negative modes. Mass range was 50–1000 m/z. An automated calibrant delivery system was used to maintain mass accuracy. Raw UPLC/QTOF MS spectral data were preprocessed using Markerview software (SCIEX, Concord, Canada) and normalized to the total spectral area. Lipids were identified using Lipid Maps (www.lipidmaps.org), Human Metabolome (www.hmdb.ca), and Metlin (metlin.scripps.edu) databases. Identification was confirmed using MS/MS pattern and retention time of lipid standards (Avanti Polar Lipids, Alabaster, USA and Sigma-Aldrich, St Louis, USA).

For lipidomic data analysis, each lipid species is indicated as a percentage of the total identified lipids. Differential-abundance analysis was performed using a two-tailed Student's t-test and presented as bar graphs or volcano plots. To determine fold-changes, each dataset was normalized, and the ratios of young vs. aged muscles, FABP3-overexpressing vs. young muscles, and FABP3-knockdown vs. aged muscles were log2-transformed and presented as volcano plots. For

correlation analysis, the log-ratio of the indicated condition for each lipid species was computed using RStudio. For principal component plots, the log-ratio of the indicated condition for each lipid species was computed using principal component analysis in Cluster 3.0[45].

**Fluorescence recovery after photobleaching (FRAP) assay**. For FRAP assay in myotubes, C2C12 myoblasts were differentiated into myotubes, and then were stained with BODIPY 500/510 C1, C12 (4,4-Difluoro-5-Methyl-4-Bora-3a,4a-Diaza-s-Indacene-3-Dodecanoic Acid) (Invitrogen) at 2 μg/ml in PBS for 10 min at 37 °C. FRAP images were acquired with a Zeiss LSM800 confocal microscope equipped with a live cell chamber (set at 37 °C or 32 °C and 5% CO$_2$) and Zen software (Zeiss). Myotubes were excited with a 488 nm laser and emission between 493 and 589 nm was recorded. Images were acquired with 16 bits image depth and 256 × 256 resolution using ~1.34 μs pixel dwell settings. Four prebleaching images were collected and then the region of interest was bleached with 70% laser power. Fluorescence intensity was measured for 5 min every 10 s. $t_{1/2}$ was defined as the time the fluorescence intensity reached 50% of maximum fluorescence recovery[46].

**Quantitative RT-PCR**. Total RNA was isolated from muscle tissues or C2C12 myotubes using RiboEX reagent (GeneAll Biotechnology Co., South Korea). Quantitative RT-PCR analysis was performed using StepOnePlus (Applied Biosystems) with a 20 μl reaction volume containing cDNA, primers and SYBR Master Mix (Applied Biosystems). The data was normalized to 36B4 mRNA expression. Quantitative RT-PCR was performed using the following primers: FABP3 sense (5′-TTC TGG AAG CTA GTG GAC AG-3′) and antisense (5′-TGA TGG TAG TAG GCT TGG TCA T-3′), Atf4 sense (5′-CCT GAA CAG CGA AGT GTT GG-3′) and antisense (5′-TGG AGA ACC CAT GAG GTT TCA A-3′) Chop sense (5′-CTG GAA GCC TGG TAT GAG GAT-3′) and antisense (5′-CAG GGT CAA GAG TAG TGA AGG T-3′), XBP1-s sense (5′-CTG AGT CCG AAT CAG GTG CAG-3′) and antisense (5′-GTC CAT GGG AAG ATG TTC TGG-3′), Grp78/Bip sense (5′-GAA AGG ATG GTT ATT GAT GCT GAG-3′) and antisense (5′-GTC TTC AAT GTC CGC ATC CTG-3′), Erdj4 sense (5′-CTT AGG TGT GCC AAA GTC TGC-3′) and antisense (5′-GGC ATC CGA GAG TGT TTC ATA-3′), Edem sense (5′-GGG ACC AAG AGG AAA AGT TTG-3′) and antisense (5′-GAG GTG AGC AGG TCA AAT CAA-3′), Degs1 sense (5′-GCC TCT GAA CTT GCT CAC CTT C-3′) and antisense (5′-TGA TCC AGG AGT TGT AGT GCG G-3′), Fads2 sense (5′-TTC CTG GAG AGC CAC TGG TTT G-3′) and antisense (5′-GAA GAA GGA CTG CTC CAC ATT GC-3′), Scd1 sense (5′-GCA AGC TCT ACA CCT GCC TCT T-3′) and antisense (5′-CGT GCC TTG TAA GTT CTG TGG C-3′), Scd2 sense (5′-GTC TGA CCT GAA AGC CGA GAA G-3′) and antisense (5′-GCA AGA AGG TGC TAA CGG CAC ACA G-3′), Elovl1 sense (5′-CTG GCT CTT CAT GCT TTC CAA GG-3′) and antisense (5′-AAG CAC CGA GTG GTG GAA GAC A-3′), Elovl4 sense (5′-ACC GTG GAG TTC TAT CGC TGG A-3′) and antisense (5′-TTT GGA ACG GCT CGC GGT CTT T-3′), Elovl5 sense (5′-GGT GGC TGT TCT TCC AGA TTG G-3′) and antisense (5′-CTT CAG GTG GTC TTT CCT CCG A-3′), Elovl6 sense (5′-CGG CAT CTG ATG AAC AAG CGA G-3′) and antisense (5′-GTA CAG CAT GTA AGC ACC AGT TC-3′), 36B4 sense (5′-AGA TTC GGG ATA TGC TGT TGG3′-) and antisense (5-′AAA GCC TGG AAG AAG GAG GTC-3′).

**Histological analysis**. TA muscle was embedded with an optimal cutting temperature (OCT) compound (Sakura Corp., Osaka, Japan) and immediately frozen in dry ice-cooled isopentane. The frozen sections were cut into 10 μm-thick cryosections with a cryostat (Thermo Electronic Corp., Waltham, MA, USA) maintained at −20 °C. Sections were washed three times with PBS and mounted. For immunofluorescence staining, serial sections were air-dried for 30 min and fixed in 4% paraformaldehyde for 30 min. Sections were washed three times with PBS, blocked in 5% BSA for 30 min and incubated with antilaminin antibody (L9393, Sigma-Aldrich) in 532 °C BSA at 4 °C overnight, followed by FITC-conjugated IgG (F-2765, Life Technologies Corp.) for 30 min. Sections were further washed in PBS and mounted with Vectashield with DAPI mounting medium (VECTOR Laboratories). Images were captured using a Nikon Eclipse Ti-U inverted microscope and Nikon DS-Ri2 camera using NIS-Elements software.

**Statistical analysis**. Fluorescence quantification and normalization were performed using ImageJ software. Data are presented as means ± S.E.M. Two-way ANOVA followed by Bonferroni post hoc test, and two-tailed unpaired Student's t-test were performed using Excel or GraphPad Prism v.8.4.2 software.

**Reporting summary**. Further information on research design is available in the Nature Research Reporting Summary linked to this article.

## Data availability

All data supporting the findings of this study are available within the paper and its supplementary information files. Source data are provided with this paper. Source data are provided with this paper.

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

## Acknowledgements

We thank Hyang Ran Yoon for FRAP assay, Seon-Kyu Kim and Ju Yeon Kwak for statistical analyses, Jae-Young Lim, Eun Soo Kwon, Jae Myoung Suh, Jeong-Yoon Kim and Yeo Jin Shin for their valuable discussion. This study was supported by grants from the Bio & Medical Technology Development Program (20110030133, 2013M3A9B6076413, 2017M3A9D8048708 K.-S.K.) of the National Research Foundation (NRF) funded by the Korean government (Ministry of Science and ICT), Korea Basic Science Institute (C060200) and the KRIBB Research Initiative Program.

## Author contributions

S.-M.L. mainly performed the study. G.-S.H. and Y.J. performed the lipidomic study. S.H.L. and J.H.Y. contributed to data acquisition. Y.L., J.Y.C., and C.Y.H. provided critical insights and resources. K.-P.L., S.S.P., and Y.H.S. performed the animal study. S.-M.L. and K.-S.K. interpreted the data. K.-S.K., S.-M.L., K.-P.L., and Y.L. wrote the manuscript. K.-S.K. conceived and supervised the study.

## Competing interests

The authors declare no competing interests.

**Additional information**

