## [Peer Review File · Nature Communications]

Reviewers' Comments:

Reviewer #1:

Remarks to the Author:

In this manuscript, Lee et al. have studied the potential role of Fatty Acid Binding Protein (FABP3) as an age-associated lipid chaperone and its implication on skeletal muscle mass and contractility. The authors demonstrate a differential lipid profile associated with FABP3 expression in adult mice compared to young. There was a substantial increase in SM and LPC lipid species as well as an increased saturation in PC and PEs of aged mice. Studies in tibialis anterior (TA) muscle of young adult mice indicate that overexpression of FABP3 iterated the lipid signature of aged mice with increased total SM, LPC, decreased unsaturation and decreased . As lipid saturation is known to increase ER stress and UPR, authors studied the involvement of FABP3 in modulating ER stress. In accordance to aged muscle, FABP3 overexpression similarly increased p-PERK, p-eIF2 α and decreased protein synthesis. Furthermore, 5 day of immobilization delayed muscle recovery and induced early fatigue in FABP3 overexpressing TA muscle. Authors further validated the role of FABP3 in the regulation of skeletal muscle physiology by inhibiting its expression in aged muscle. FABP3-knockdown decreased the levels of SM and LPC and indicated inverse correlation with FABP3-overexpressing as well as aged mice. They observed an increase in lipid unsaturation which was similar to young mice comprehensively supporting the role of FABP3 in age- dependent lipid remodeling. Moreover FABP3 inhibition also reduced p-PERK, p eIF2 α , improved protein synthesis and increased membrane fluidity. Addition of PUFAs in FABP3 overexpressing C2C12 myotubes was also able to alleviate ER stress. Authors demonstrated that FABP3 knockdown slows down muscle loss due to immobilization and increased it during remobilization. The manuscript is well written, thoughtfully conceived and supported by robust experimental data. However, there are several issues which authors should address in the manuscript.

Specific Comments:

1. The authors should investigate the protein levels/activities of fatty acid desaturases and elongases in myotubes or tissues of young, old, FABP3 over-expressed and knocked-down mice. Although, the authors referred to articles studying possible association of desaturase and elongases yet it would be intriguing to study the role of FABP3 on fatty acid synthesizing enzymes.
2. Authors suggest that increased pPERK, p-eIF2 α and decrease in protein synthesis is independent of mTOR pathway. First of all the data in supplemental figure about mTOR is not convincing. Authors should provide quantification from multiple mice in each group. PERK overexpression is known to inhibit protein synthesis by inhibiting 43S pre-initiation complex (Miyake M et al. Ligand-induced rapid skeletal muscle atrophy in HSA-Fv2E-PERK transgenic mice. PLoS ONE 12 (6): e0179955. (2017). It would be intriguing to investigate whether FABP3-mediated PERK upregulation inhibits protein synthesis through this axis.
3. Overexpression of FABP3 in young mice led to a change in PERK pathway proteins. Authors should provide the time point at which the tissues were collected.
4. Authors should discuss the rationale for studying the immobilization-remobilization model.
5. Although supplementing DHA restores membrane fluidity and decrease pPERK and peIF2 α in FABP3 overexpressing myotubes yet some indications of improved myotubes health should be provided. Myotube pictographs under various conditions can be useful.
6. As in the case of FABP3 overexpression, authors should include data showing the effects of FABP3 knockdown on muscle physiology of aged mice in parallel with the PERK and eIF2 α levels. In the immobilization and remobilization model of FABP3 knocked down mice the improved muscle function should also be reported along with PERK and eIF2 α levels.

Reviewer #2:

Remarks to the Author:

The manuscript describes the involvement of FABP3 on ER stress-mediated skeletal muscle aging by modulating membrane lipid composition and fluidity. This is potentially interesting, but there are several concerns that should be addressed. My concerns include the following.

Comments

1. Lipidomic analyses demonstrated the change of lipid composition by overexpressed and knockdown of FABP3. FABP3 is a fatty acid-binding protein. The ligand of FABP3 may play significant roles of modulating membrane lipid composition and fluidity. However, it is unclear which kind of fatty acids binds to FABP3 in these setups. This should be critically addressed.
2. ER stress pathways includes not only PERK-eIF2a pathway but also ATF6-mediated chaperone induction and IRE1-mediated several pathways including ASK1-JNK, NF- κ B, and XBP1 splicing. All pathways should be addressed if "ER stress"-mediated action is mentioned.
3. P-eIF2a is time-dependently regulated in a usual condition. Detailed methods should be shown.
4. It has recently been reported that FABP4, another FABP, derived from mainly adipocytes is secreted from cells and acts as an adipokine. Is FABP3 secreted from muscle cells? Does FABP3 have similar effects as a bioactive molecule?
5. What is the mechanism or reason of the differential regulation of FABP3 between skeletal muscle and heart?
6. What is the mechanism of increase in FABP3 by aging? This point is totally unclear in this study.

Reviewer #3:

Remarks to the Author:

To the authors

The present paper links the expression of FABP3 in aged skeletal muscle with alterations in lipid composition, ER stress signaling, and muscle protein synthesis and muscle contractility. The authors have gone to great lengths to examine the role of FABP3 in this respect. Knockdown and overexpression of FABP3 in in vitro and in vivo models have been employed and appear to support a role of FABP3 in augmenting lysophosphatidylcholine species and sphingolipids, along with increased lipid saturation, reduced membrane fluidity, and induction of ER stress antecedent to downregulation of muscle protein synthesis. This has led the authors to (somewhat prematurely in my opinion) conclude that increased FABP3 in aged skeletal muscle may represent a therapeutic target for prevention of age-associated sarcopenia.

This paper comprises a set of a well-designed and well-executed experiments demonstrating a link between FABP3 and ER stress induction, and altered lipid composition in aged skeletal muscle and the upregulation of FABP3 in aging is a commendable novel finding in this research, however, the mechanistic conclusions are tenuous at best and are not fully elucidated in the manuscript. There is little analysis on how these lipid species may result in ER stress and the suggested mechanism related to membrane fluidity is correlative and speculative. No mechanistic link between ER stress and membrane fluidity has been presented. The authors are encouraged to elaborate on this in a revised version of their paper.

Several alternative mechanisms that could link ER stress to membrane fluidity could be at play in this context which deserve further exploration (alternatively, the authors may consider toning down some of the rather firm conclusions drawn. For example, increases in ceramides have been linked to reduced insulin signaling and ER stress induction in skeletal muscle. Moreover, ceramides are notably hydrophobic in comparison to other sphingolipids and therefore would be more susceptible to changes in lipid chaperoning. Furthermore, reduced insulin signaling through Akt

could also play a role in reduced muscle protein synthesis, and this has not been entirely ruled out by the lack of change in mTOR signaling. Indeed, only 1 arm of the downstream signaling for mTOR has been measured. The authors could consider including an assessment of p70S6k expression and Akt phosphorylation to strengthen this aspect of their conclusions. Also given that ceramides are known to act through PKC signaling, this is also a target that could be considered for further analysis. While the authors also present some good data showing reduced membrane fluidity in FABP3 overexpression, it is not clear that this results in the induction of ER-stress and this aspect should be suggested only as a possible mechanistic link, and therefore removed from the title, unless stronger mechanistic support for this hypothesis can be provided. Moreover, although membrane fluidity was altered the authors did not show any data relating to the composition of the plasma, or intracellular membranes.

I disagree with the concluding sentence of the abstract, which clearly is an overstatement of what has been shown. The concluding sentence of the discussion already is more appropriate as that indeed only mentions one target (being FABP3), rather than an range of targets/pathways, which are unlikely to be selectively hit by one single therapeutic intervention.

- Validity:

The main flaws of the manuscript are an incomplete and hence biased assessment of the potential signaling pathways that could mediate the functional outcomes along with incomplete mechanistic support for the stated conclusions.

- Originality and significance:

The identity of FABP3 as a possible target for age associated sarcopenia is novel and interesting and warrants further investigation.

- Data & methodology: No major issues here

- Appropriate use of statistics and treatment of uncertainties: No major issues here

- Conclusions: Do you find that the conclusions and data interpretation are robust, valid and reliable?

Although the main conclusions are reasonably well supported, the authors have overstated some of the possible mechanistic links.

- Suggested improvements: Please list additional experiments or data that could help strengthening the work in a revision.

See in the section 'To the authors' .

- References: Does this manuscript reference previous literature appropriately? If not, what references should be included or excluded?

No problems here

- Clarity and context: Is the abstract clear, accessible? Are abstract, introduction and conclusions appropriate?

No major issues here, except for the strong link that is made that 'FABP3-lipid composition-membrane fluidity-ER stress axis is a target for sarcopenia treatment'. Basically this is not a single target but rather quite a range of diverse mechanisms which indeed all are likely to affect sarcopenia, but which can be hardly viewed as a single target for treatment.

Reviewers' comments:

Reviewer #1 (Remarks to the Author):

In this manuscript, Lee et al. have studied the potential role of Fatty Acid Binding Protein (FABP3) as an age-associated lipid chaperone and its implication on skeletal muscle mass and contractility. The authors demonstrate a differential lipid profile associated with FABP3 expression in adult mice compared to young. There was a substantial increase in SM and LPC lipid species as well as an increased saturation in PC and PEs of aged mice. Studies in tibialis anterior (TA) muscle of young adult mice indicate that overexpression of FABP3 iterated the lipid signature of aged mice with increased total SM, LPC, decreased unsaturation and decreased . As lipid saturation is known to increase ER stress and UPR, authors studied the involvement of FABP3 in modulating ER stress. In accordance to aged muscle, FABP3 overexpression similarly increased p-PERK, p-eIF2 α and decreased protein synthesis. Furthermore, 5 day of immobilization delayed muscle recovery and induced early fatigue in FABP3 overexpressing TA muscle. Authors further validated the role of FABP3 in the regulation of skeletal muscle physiology by inhibiting its expression in aged muscle. FABP3-knockdown decreased the levels of SM and LPC and indicated inverse correlation with FABP3-overexpressing as well as aged mice. They observed an increase in lipid unsaturation which was similar to young mice comprehensively supporting the role of FABP3 in age-dependent lipid remodeling. Moreover FABP3 inhibition also reduced p-PERK, p eIF2 α , improved protein synthesis and increased membrane fluidity. Addition of PUFAs in FABP3 overexpressing C2C12 myotubes was also able to alleviate ER stress. Authors demonstrated that FABP3 knockdown slows down muscle loss due to immobilization and increased it during remobilization.

The manuscript is well written, thoughtfully conceived and supported by robust experimental data. However, there are several issues which authors should address in the manuscript.

Specific Comments:

1. The authors should investigate the protein levels/activities of fatty acid desaturases and elongases in myotubes or tissues of young, old, FABP3 over-expressed and knocked-down mice. Although, the authors referred to articles studying possible association of desaturase and elongases yet it would be intriguing to study the role of FABP3 on fatty acid synthesizing enzymes.
2. Authors suggest that increased pPERK, p-eIF2 α and decrease in protein synthesis is independent of mTOR pathway. First of all the data in supplemental figure about mTOR is not convincing. Authors should provide quantification from multiple mice in each group. PERK overexpression is known to inhibit protein synthesis by inhibiting 43S pre-initiation complex (Miyake M et al. Ligand-induced rapid skeletal muscle atrophy in HSA-Fv2E-PERK transgenic mice. PLoS ONE 12 (6): e0179955. (2017). It would be intriguing to investigate whether FABP3-mediated PERK upregulation inhibits protein synthesis through this axis.
3. Overexpression of FABP3 in young mice led to a change in PERK pathway proteins. Authors should provide the time point at which the tissues were collected.

4. Authors should discuss the rationale for studying the immobilization-remobilization model.

5. Although supplementing DHA restores membrane fluidity and decrease pPERK and p-eIF2 α in FABP3 overexpressing myotubes yet some indications of improved myotubes health should be provided. Myotube pictographs under various conditions can be useful.

6. As in the case of FABP3 overexpression, authors should include data showing the effects of FABP3 knockdown on muscle physiology of aged mice in parallel with the PERK and eIF2 α levels. In the immobilization and remobilization model of FABP3 knocked down mice the improved muscle function should also be reported along with PERK and eIF2 α levels.

Reviewer #2 (Remarks to the Author):

The manuscript describes the involvement of FABP3 on ER stress-mediated skeletal muscle aging by modulating membrane lipid composition and fluidity. This is potentially interesting, but there are several concerns that should be addressed. My concerns include the following.

Comments

1. Lipidomic analyses demonstrated the change of lipid composition by overexpressed and knockdown of FABP3. FABP3 is a fatty acid-binding protein. The ligand of FABP3 may play significant roles of modulating membrane lipid composition and fluidity. However, it is unclear which kind of fatty acids binds to FABP3 in these setups. This should be critically addressed.
2. ER stress pathways includes not only PERK-eIF2 α pathway but also ATF6-mediated chaperone induction and IRE1-mediated several pathways including ASK1-JNK, NF- κ B, and XBP1 splicing. All pathways should be addressed if “ER stress”-mediated action is mentioned.
3. P-eIF2 α is time-dependently regulated in a usual condition. Detailed methods should be shown.
4. It has recently been reported that FABP4, another FABP, derived from mainly adipocytes is secreted from cells and acts as an adipokine. Is FABP3 secreted from muscle cells? Does FABP3 have similar effects as a bioactive molecule?
5. What is the mechanism or reason of the differential regulation of FABP3 between skeletal muscle and heart?
6. What is the mechanism of increase in FABP3 by aging? This point is totally unclear in this study.

Reviewer #3 (Remarks to the Author):

To the authors

The present paper links the expression of FABP3 in aged skeletal muscle with alterations in lipid composition, ER stress signaling, and muscle protein synthesis and muscle contractility. The authors have gone to great lengths to examine the role of FABP3 in this respect. Knockdown and overexpression of FABP3 in in vitro and in vivo models have been

employed and appear to support a role of FABP3 in augmenting lysophosphatidylcholine species and sphingolipids, along with increased lipid saturation, reduced membrane fluidity, and induction of ER stress antecedent to downregulation of muscle protein synthesis. This has led the authors to (somewhat prematurely in my opinion) conclude that increased FABP3 in aged skeletal muscle may represent a therapeutic target for prevention of age-associated sarcopenia.

This paper comprises a set of a well-designed and well-executed experiments demonstrating a link between FABP3 and ER stress induction, and altered lipid composition in aged skeletal muscle and the upregulation of FABP3 in aging is a commendable novel finding in this research, however, the mechanistic conclusions are tenuous at best and are not fully elucidated in the manuscript. There is little analysis on how these lipid species may result in ER stress and the suggested mechanism related to membrane fluidity is correlative and speculative. No mechanistic link between ER stress and membrane fluidity has been presented. The authors are encouraged to elaborate on this in a revised version of their paper.

Several alternative mechanisms that could link ER stress to membrane fluidity could be at play in this context which deserve further exploration (alternatively, the authors may consider toning down some of the rather firm conclusions drawn. For example, increases in ceramides have been linked to reduced insulin signaling and ER stress induction in skeletal muscle. Moreover, ceramides are notably hydrophobic in comparison to other sphingolipids and therefore would be more susceptible to changes in lipid chaperoning. Furthermore, reduced insulin signaling through Akt could also play a role in reduced muscle protein synthesis, and this has not been entirely ruled out by the lack of change in mTOR signaling. Indeed, only 1 arm of the downstream signaling for mTOR has been measured. The authors could consider including an assessment of p70S6k expression and Akt phosphorylation to strengthen this aspect of their conclusions. Also given that ceramides are known to act through PKC signaling, this is also a target that could be considered for further analysis. While the authors also present some good data showing reduced membrane fluidity in FABP3 overexpression, it is not clear that this results in the induction of ER-stress and this aspect should be suggested only as a possible mechanistic link, and therefore removed from the title, unless stronger mechanistic support for this hypothesis can be provided. Moreover, although membrane fluidity was altered the authors did not show any data relating to the composition of the plasma, or intracellular membranes.

I disagree with the concluding sentence of the abstract, which clearly is an overstatement of what has been shown. The concluding sentence of the discussion already is more appropriate as that indeed only mentions one target (being FABP3), rather than an range of targets/pathways, which are unlikely to be selectively hit by one single therapeutic intervention.

- Validity:

The main flaws of the manuscript are an incomplete and hence biased assessment of the potential signaling pathways that could mediate the functional outcomes along with incomplete mechanistic support for the stated conclusions.

- Originality and significance:

The identity of FABP3 as a possible target for age associated sarcopenia is novel and interesting and warrants further investigation.

- Data & methodology: No major issues here
- Appropriate use of statistics and treatment of uncertainties: No major issues here
- Conclusions: Do you find that the conclusions and data interpretation are robust, valid and reliable?
Although the main conclusions are reasonably well supported, the authors have overstated some of the possible mechanistic links.
- Suggested improvements: Please list additional experiments or data that could help strengthening the work in a revision.
See in the section 'To the authors' .
- References: Does this manuscript reference previous literature appropriately? If not, what references should be included or excluded?
No problems here
- Clarity and context: Is the abstract clear, accessible? Are abstract, introduction and conclusions appropriate?
No major issues here, except for the strong link that is made that 'FABP3-lipid composition-membrane fluidity-ER stress axis is a target for sarcopenia treatment'. Basically this is not a single target but rather quite a range of diverse mechanisms which indeed all are likely to affect sarcopenia, but which can be hardly viewed as a single target for treatment.

Point-by-point responses are as follows,

Reviewers' comments:

Reviewer #1 (Remarks to the Author):

In this manuscript, Lee et al. have studied the potential role of Fatty Acid Binding Protein (FABP3) as an age-associated lipid chaperone and its implication on skeletal muscle mass and contractility. The authors demonstrate a differential lipid profile associated with FABP3 expression in adult mice compared to young. There was a substantial increase in SM and LPC lipid species as well as an increased saturation in PC and PEs of aged mice. Studies in tibialis anterior (TA) muscle of young adult mice indicate that overexpression of FABP3 iterated the lipid signature of aged mice with increased total SM, LPC, decreased unsaturation and decreased . As lipid saturation is known to increase ER stress and UPR, authors studied the involvement of FABP3 in modulating ER stress. In accordance to aged muscle, FABP3 overexpression similarly increased p-PERK, p-eIF2 α and decreased protein synthesis. Furthermore, 5 day of immobilization delayed muscle recovery and induced early fatigue in FABP3 overexpressing TA muscle. Authors further validated the role of FABP3 in the regulation of skeletal muscle physiology by inhibiting its expression in aged muscle. FABP3-knockdown decreased the levels of SM and LPC and indicated inverse correlation with FABP3-overexpressing as well as aged mice. They observed an increase in lipid unsaturation which was similar to young mice comprehensively supporting the role of FABP3 in age-dependent lipid remodeling. Moreover FABP3 inhibition also reduced p-PERK, p eIF2 α , improved protein synthesis and increased membrane fluidity. Addition of PUFAs in FABP3 overexpressing C2C12 myotubes was also able to alleviate ER stress. Authors demonstrated that FABP3 knockdown slows down muscle loss due to immobilization and increased it during remobilization.

The manuscript is well written, thoughtfully conceived and supported by robust experimental data. However, there are several issues which authors should address in the manuscript.

Specific Comments:

Comment

1. The author should investigate the protein levels/activities of fatty acid desaturases and elongases in myotubes or tissues of young, old, FABP3 over-expressed and knocked-down mice. Although, the authors referred to articles studying possible association of desaturase and elongases yet it would be intriguing to study the role of FABP3 on fatty acid synthesizing enzymes.

We appreciate this suggestion and in response, measured the mRNA expression levels of fatty acid synthesizing enzymes. Since FABPs are known to escort lipids to different cellular compartments rather than directly regulate the protein levels/activities of fatty acid desaturases and elongases (Furuhashi and Hotamisligil 2008), and FABP4 is known to regulate mRNA expression of *Scd1* (stearoyl-CoA desaturase-1) (Erbay, Babaev et al. 2009), we measured mRNA expression of fatty acid desaturases such as *Degs*, *Fads2*, *Scd1* and *Scd2* as well as skeletal muscle specific elongases such as *Elovl1*, *Elovl4*, *Elovl5* and *Elovl6*. We did not observe any significant differences in mRNA expression levels of these genes in tissues of young, old, FABP3 over-expressed, or knocked-down mice (new Supplementary Fig. 6 a, b). These results suggest that FABP3 alters membrane lipid composition by directly

delivering SFAs to the membrane, rather than by regulating desaturase and elongase gene expression. We have added Supplemental Fig. 6a and b as well as a corresponding description of these results in the Discussion section of the revised manuscript (page 16, line 15).

2. Authors suggest that increased pPERK, p-eIF2 α and decrease in protein synthesis is independent of mTOR pathway. First of all the data in supplemental figure about mTOR is not convincing. Authors should provide quantification from multiple mice in each group. PERK overexpression is known to inhibit protein synthesis by inhibiting 43S pre-initiation complex (Miyake M et al. Ligand-induced rapid skeletal muscle atrophy in HSA-Fv2E-PERK transgenic mice. PLoS ONE 12 (6): e0179955. (2017). It would be intriguing to investigate whether FABP3-mediated PERK upregulation inhibits protein synthesis through this axis.

In response to the reviewer's suggestion, we have examined phosphorylation of 4EBP1 and S6K in FABP3 overexpressing C2C12 myotubes (revised Supplemental Fig. 3g) to confirm mTOR dependency. In addition, we examined phosphorylation of mTOR, S6K, and 4EBP1 in TA muscle (n = 3) transfected with FABP3 or vehicle. Consistent with the previous mTOR blot, we did not observe significant effects of FABP3 overexpression on phosphorylation of mTOR, S6K, or 4EBP1 (new Supplemental Fig. 3h). This result indicates that FABP3 inhibits *de novo* protein synthesis in an mTOR-independent manner. We have also provided quantification of the western blot (Supplementary Fig. 3g, h).

Furthermore, as noted by the reviewer, PERK activation has been shown to increase eIF2 α phosphorylation, which in turn inhibits formation of the 43S pre-initiation complex, thereby leading to defective protein synthesis (Krishnamoorthy, Pavitt et al. 2001, Miyake, Kuroda et al. 2017). Since FABP3 induced phosphorylation of both PERK and eIF2 α , and contributes to reduced protein synthesis in muscles, we agree that FABP3-mediated ER stress is likely to inhibit *de novo* protein synthesis through the PERK-eIF2 α -43S pre-initiation complex axis. To investigate whether FABP3-induced inhibition of protein synthesis depends on PERK activation, we inhibited PERK activity by GSK2606414 and found that FABP3-induced inhibition of protein synthesis was almost totally rescued (new Supplementary Fig. 3c). Thus, we have addressed these in the Results section of the revised manuscript (page 8, line 23 and page 9, line 3).

3. Overexpression of FABP3 in young mice led to a change in PERK pathway proteins. Author should provide the time point at which the tissues were collected.

The time point has been updated in the relevant Fig.3b legends of the revised manuscript (page 38, line 8).

4. Authors should discuss the rationale for studying the immobilization-remobilization model.

We appreciate the request to further explain our motive for this study. Aging is accompanied by impaired muscle recovery, resulting in accelerated muscle mass loss (White, Confides et al. 2015). It has been reported that the muscle atrophy occurring during immobilization is similar between young and aged animals (Siu and Alway 2005), whereas recovery after immobilization is attenuated in aged animals compared to young animals (Degens and Alway

2003, Suetta, Clemmensen et al. 2010). Furthermore, immobilization is a catabolic process that induces muscle atrophy with increased atrogin-1 and MuRF1 expression, which are muscle-specific ubiquitin ligases (Krawiec, Frost et al. 2005). Remobilization is an anabolic process that induces regrowth with increases protein synthesis (Childs, Spangenburg et al. 2003, Slimani, Vazeille et al. 2015). We, therefore, examined the effect of FABP3 on muscle atrophy and muscle recovery during immobilization and remobilization by measuring muscle mass and force according to previously published protocols (Caron, Drouin et al. 2009). We found that FABP3 inhibited only muscle mass recovery during the remobilization period without increasing the degree of muscle loss during immobilization (Fig.4b). This may be due to FABP3 blocking protein synthesis pathways that are crucial for the recovery process. As requested by the reviewer, we have added this point to the Results section of the revised manuscript (page 10, line 19).

5. Although supplementing DHA restores membrane fluidity and decrease pPERK and pEIF2 α in FABP3 overexpressing myotubes yet some indications of improved myotubes health should be provided. Myotube pictographs under various conditions can be useful.

We thank the reviewer for this valuable suggestion. We have added myotube morphology data showing the effect of supplementing DHA on myotube recovery after dexamethasone-induced atrophy. Consistent with our previous results, which showed that supplementing DHA restored membrane fluidity and reduction of PERK and eIF2 α phosphorylation (revised Fig 7a, b), DHA supplementation increased myotube diameter (new Fig.7c). We have added myotube images and pictographs of digitizing myotube diameters, as well as a description in the Results section of the revised manuscript (page 14, line 2).

6. As in the case of FABP3 overexpression, authors should include data showing the effects of FABP3 knockdown on muscle physiology of aged mice in parallel with the PERK and eIF2 α levels. In the immobilization and remobilization model of FABP3 knocked down mice the improved muscle function should also be reported along with PERK and eIF2 α levels.

In contrast to FABP3 overexpression (new Fig. 4e) (page 11, line 2), FABP3 knockdown decreased phosphorylation of both PERK and eIF2 α (new Fig. 8e) (page 14, line 16), while improving recovery of muscle mass and strength (revised Fig. 8b-i) after remobilization.

Reviewer #2 (Remarks to the Author):

The manuscript describes the involvement of FABP3 on ER stress-mediated skeletal muscle aging by modulating membrane lipid composition and fluidity. This is potentially interesting, but there are several concerns that should be addressed. My concerns include the following.

Comments

1. Lipidomic analyses demonstrated the change of lipid composition by overexpressed and knockdown of FABP3. FABP3 is a fatty acid-binding protein. The ligand of FABP3 may play significant roles of modulating membrane lipid composition and fluidity. However, it is unclear which kind of fatty acids binds to FABP3 in these setups. This should be critically addressed.

We appreciate the reviewer's comments and agree that we did not provide sufficient details regarding the types of fatty acids that bind to FABP3 in muscle of aged mice. It is known that the dissociation constants (K_d) of FABP3 for saturated fatty acids (SFA) such as palmitate and stearate are 14, and 4, respectively, whereas the K_d for polyunsaturated fatty acids (PUFA) such as linoleate, linolenate, and arachidonate are 32, 38, and 39 nM, respectively (Richieri, Ogata et al. 1994). Therefore, we supposed that FABP3 preferentially binds to SFA with higher affinity than PUFA, thereby leading to membrane lipid saturation. We have addressed this in the Discussion section of the revised manuscript (page 16, line 17).

2. ER stress pathways includes not only PERK-eIF2a pathway but also ATF6-mediated chaperone induction and IRE1-mediated several pathways including ASK1-JNK, NF- κ B, and XBP1 splicing. All pathways should be addressed if "ER stress"-mediated action is mentioned.

Thank you for this suggestion. Accordingly, we investigated whether other ER membrane-associated sensors including IRE-1 α and ATF6 could be involved in FABP3-driven ER stress (Afroze and Kumar 2019). We found that FABP3 did not induce phosphorylation of IRE-1 α or its downstream effectors, JNK, p65, and SEK, nor did it induce ATF6 cleavage (revised Supplementary Fig. 3d, f). We also did not observe significant changes in mRNA expression levels of unfolded protein response genes, *Xbp1-s*, *Grp78/Bip*, *Erdj4*, and *Edem*, (Supplementary Fig. 3e). Therefore, we suggest that FABP3 induces ER stress primarily through the PERK-eIF2 α pathway. We have added the corresponding description to the Results section of the revised manuscript (page 9 line 7).

3. P-eIF2a is time-dependently regulated in a usual condition. Detailed methods should be shown.

As requested by the reviewer, we monitored eIF2 α phosphorylation over time after FABP3 expression. To this end, we established a stable C2C12 cell line that overexpressed FABP3 under the control of Cre recombinase. The stable cells were differentiated into myotubes for 3 days, after which, the fully differentiated C2C12 myotubes were infected with Cre-carrying adenovirus (Ad-Cre). FABP3 expression was increased in differentiated C2C12 myotubes at 24 h after infection and maintained for 72 h. The level of PERK and eIF2 α phosphorylation gradually increased in FABP3-overexpressing myotubes at 36 h post-infection until 72 h. Further, the *de novo* protein synthesis was decreased from 36 h after infection. We have added Supplemental Fig. 3a and revised the corresponding description in the Results section and the corresponding figure legend of the revised manuscript (page 8, line 22 and page 54, line 5).

4.1. It has recently been reported that FABP4, another FABP, derived from mainly adipocytes is secreted from cells and acts as an adipokine. Is FABP3 secreted from muscle cells?

We thank the reviewer for this important question. It has been reported that FABP3 is a valuable serum biomarker of heart and skeletal muscle injury (Pelsers, Hermens et al. 2005, Pritt, Hall et al. 2008). We explored whether the circulating FABP3 level changes with age. Surprisingly, we found that the circulating FABP3 level was significantly elevated in aged individuals compared with young individuals (4.432 vs. 2.308 ng/mL, p-value = 0.0013) (see

figure below for reviewers). As we found that FABP3 expression was increased in skeletal muscles and not in the heart with age (Fig. 1a), we anticipate that the origin of circulating FABP3 is skeletal muscle. However, we do not yet know whether the circulating FABP3 is actively or passively secreted from skeletal muscle. This data have not been included in the manuscript, because it requires further research.

Comparison of serum FABP3 levels between young and aged human subjects.

Participants were healthy elderly (age ≥ 65 years) and healthy young adults (20-40 years) without uncontrolled chronic diseases. The protocol was approved by the institutional review board of SNUBH (B-1307-212-008). The circulating FABP3 levels were determined using ELISA for ten samples from each age group.

4.2. Does FABP3 have similar effects as a bioactive molecule?

As mentioned by the reviewer, it has been reported that secreted FABP4 from adipocytes regulates hepatic glucose production (Cao, Sekiya et al. 2013). However, as far as we know, there is no evidence to suggest that secreted FABP3, a lipid chaperone, acts as a myokine to transfer autocrine or paracrine signaling communication. If FABP3 is a novel myokine that acts not only in the skeletal muscle but also in other tissues (or organs), it might be a more attractive target for therapeutic interventions of age-related diseases.

5. What is the mechanism or reason of the differential regulation of FABP3 between skeletal muscle and heart?

We are grateful for the reviewer's question. We are also very interested in tissue specific regulatory mechanism of FABP3 expression and plan to explore this in the future. Interestingly, it has been reported that skeletal muscle and heart show distinct aging phenotypes: atrophy in aging skeletal muscle vs. hypertrophy in aging heart tissue (Park and Prolla 2005). Furthermore, the results of gene expression analysis (GSE11292) showed that the age-specific gene expression profile of skeletal muscle differed significantly from that of the heart (see figure below for reviewers). In fact, a total of 98 genes were differentially expressed between skeletal muscle of young (5 months) and old (30 months) mice ($|\log_2FC| \geq 1.5$, adj. p-value < 0.1), which included 44 up-regulated and 54 down-regulated genes (left panel). However, no significant changes in the 98 differentially expressed genes were observed in the heart (right panel), indicating that these two tissues have distinct gene regulatory mechanisms in their aging processes. It is, therefore, not surprising that expression of FABP3 is differentially controlled in skeletal muscle and heart.

Age-specific gene expression profile differ markedly between skeletal muscle and heart.

6. What is the mechanism of increase in FABP3 by aging? This point is totally unclear in this study.

We appreciate this question as we are also very interested in the molecular mechanism associated with the age-related increase in FABP3 expression in skeletal muscle and, as such, have been continuing to study this. Based on our findings thus far, we suggest that age-dependent decrease in SIRT1 activity induces FABP3 expression in skeletal muscle and contributes to sarcopenia. It has been well documented that SIRT1 activity is dependent on NAD⁺ levels, which decline with age in skeletal muscle (Gomes, Price et al. 2013). We, therefore, investigated the effect of SIRT1 inhibition by gene silencing, or treatment with chemical inhibitors, on FABP3 levels and found that SIRT1 inhibition increased both protein and mRNA levels of FABP3 (see figure below for reviewers). Therefore, we speculate that decreasing activity of SIRT1 during aging gradually increases FABP3 expression in the skeletal muscle, which is a hypothesis that requires further investigations to elucidate the precise molecular mechanism. We have addressed this in the Discussion section of the revised manuscript (page 18, line 11)

SIRT1 inhibition increases FABP3 expression in C2C12 myotubes. (a, b) C2C12 myotubes were transfected with siSirt1 or siControl, after which protein and mRNA levels were analyzed at 48 h post-transfection. (c) C2C12 myotubes were treated with SIRT1 inhibitors, EX-527 (25 μ M) or sirtinol (5 μ M) for 48 h. Data are presented as means \pm S.E.M. *** $p < 0.001$.

Reviewer #3 (Remarks to the Author):

To the authors

The present paper links the expression of FABP3 in aged skeletal muscle with alterations in lipid composition, ER stress signaling, and muscle protein synthesis and muscle contractility. The authors have gone to great lengths to examine the role of FABP3 in this respect. Knockdown and overexpression of FABP3 in in vitro and in vivo models have been employed and appear to support a role of FABP3 in augmenting lysophosphatidylcholine species and sphingolipids, along with increased lipid saturation, reduced membrane fluidity, and induction of ER stress antecedent to downregulation of muscle protein synthesis. This has led the authors to (somewhat prematurely in my opinion) conclude that increased FABP3 in aged skeletal muscle may represent a therapeutic target for prevention of age-associated sarcopenia.

This paper comprises a set of a well-designed and well-executed experiments demonstrating a link between FABP3 and ER stress induction, and altered lipid composition in aged skeletal muscle and the upregulation of FABP3 in aging is a commendable novel finding in this research, however, the mechanistic conclusions are tenuous at best and are not fully elucidated in the manuscript. There is little analysis on how these lipid species may result in ER stress and the suggested mechanism related to membrane fluidity is correlative and speculative. No mechanistic link between ER stress and membrane fluidity has been presented. The authors are encouraged to elaborate on this in a revised version of their paper.

We appreciate the reviewer's comments and in response, we investigated whether membrane fluidity is associated with ER stress. Membranes become more rigid when either the saturated lipid content increases or the temperature decreases (Fan and Evans 2015). Therefore, after decreasing the temperature from 37 $^{\circ}$ C to 32 $^{\circ}$ C, we performed FRAP analysis to monitor membrane fluidity, and western blot analysis to detect ER stress. As expected, membrane fluidity was decreased at the low temperature (open circle vs. grey circle in new Fig. 3f). Moreover, we found that the low temperature markedly aggravated membrane fluidity in FABP3-overexpressing myotubes (light pink circle vs. red circle in new Fig. 3f) and resulted in a more severe ER stress response (PERK and eIF2 α phosphorylation) and defective protein synthesis (new Fig. 3g). The above results were in concordance with our previous results, which showed that supplementing DHA (22:6, polyunsaturated fatty acids) increased membrane fluidity, and consequently mitigated FABP3-induced ER stress (revised Fig. 7a, b).

These results indicate that fluidity changes due to temperature (a physical element) and lipid composition (a chemical element) function together to induce ER stress, and therefore support, at least in part, our suggested mechanism that FABP3-induced alteration in membrane fluidity promotes ER stress. We have added the corresponding description in the Results sections of the revised manuscript (page 10, line 7).

Several alternative mechanisms that could link ER stress to membrane fluidity could be at play in this context which deserve further exploration (alternatively, the authors may consider toning down some of the rather firm conclusions drawn. For example, increases in ceramides have been linked to reduced insulin signaling and ER stress induction in skeletal muscle. Moreover, ceramides are notably hydrophobic in comparison to other sphingolipids and therefore would be more susceptible to changes in lipid chaperoning. Furthermore, reduced insulin signaling through Akt could also play a role in reduced muscle protein synthesis, and this has not been entirely ruled out by the lack of change in mTOR signaling. Indeed, only 1 arm of the downstream signaling for mTOR has been measured. The authors could consider including an assessment of p70S6k expression and Akt phosphorylation to strengthen this aspect of their conclusions. Also given that ceramides are known to act through PKC signaling, this is also a target that could be considered for further analysis.

We appreciate this alternative suggestion and investigated a possible association between reduced insulin signaling and ER stress by measuring phosphorylation of Akt and p70S6K, as well as PKC signaling in FABP3-overexpressing myotubes. FABP3 overexpression did not induce phosphorylation of AKT, p70S6K or GSK-3 β (Supplementary Fig. 3g). Moreover, FABP3 overexpression did not affect PKC ζ downstream such as SEK and JNK phosphorylation (Bourbon, Yun et al. 2000) (Supplementary Fig. 3f). Together, these results indicate that insulin signaling is not implicated in defective protein synthesis induced by FABP3-mediated ER stress. Moreover, since the content of ceramide is as low as 0.16 % in young, and 0.29% in old muscles, it is unlikely that ceramide plays a major role in FABP3-mediated ER stress and defective protein synthesis (Supplementary Table).

While the authors also present some good data showing reduced membrane fluidity in FABP3 overexpression, it is not clear that this results in the induction of ER-stress and this aspect should be suggested only as a possible mechanistic link, and therefore removed from the title, unless stronger mechanistic support for this hypothesis can be provided. Moreover, although membrane fluidity was altered the authors did not show any data relating to the composition of the plasma, or intracellular membranes.

Thank you for this comment. Accordingly, we added data in which a physical decrease in membrane fluidity at low temperatures aggravate the FABP3-induced ER stress, while increased membrane fluidity by DHA supplementation alleviates the FABP3-induced ER stress. Although these results demonstrate that membrane fluidity is associated with ER stress, we do not yet understand how altering the membrane fluidity induces ER stress mechanistically. We have described possible mechanisms in the Discussion section and modified our title; “FABP3-mediated membrane lipid saturation alters fluidity and induces ER stress in skeletal muscle with aging” (page 1, line 1)

This reviewer is correct in pointing out that we did not investigate separately compositional changes in the plasma membrane and intracellular membrane. Nonetheless, there are specific reasons as to why we considered the observed alteration in membrane composition due to FABP3 or age, as the result of changes in the ER membrane. First, the ER membrane typically constitutes more than 50% of the total membrane of an average animal cell (Alberts, Johnson et al. 2002). Second, especially in skeletal muscle, the ER membrane constitutes more than that in other cells and tissue, as muscles forms through the fusion of large numbers of myoblasts into multi-nucleated myotubes (Kim, Jin et al. 2015) and, therefore, the ratio of plasma membrane to intracellular ER membrane decreases by several folds. Consistently, the composition that we identified here was similar to that of ER. Our lipidomic analysis showed that phosphatidylcholine (PC), sphingomyelin (SM), and lysophosphatidylcholine (LPC) account for 76%, 2.91%, and 1.13%, respectively, of the total identified lipid. Compared with the membrane compositions of Golgi (PC: 45.3%, SM: 12.3%, and LPC: 5.9%) and plasma (PC: 43.1%, SM: 23.1%, and LPC:1.8%), our data showed the highest PC and lowest SM content that is typical in ER (PC: 54.4 - 59.6, SM: 2.4 - 6.3%, and LPC: 2.9%) (Vance and Vance 2008). For the above stated reasons, we believe that the result of our lipidomic analysis are primarily attributed to the altered ER membrane lipid composition. However, we cannot exclude the possibility that FABP3-driven membrane lipid saturation affects other membranes' functions, and thus we plan to explore this in future work.

I disagree with the concluding sentence of the abstract, which clearly is an overstatement of what has been shown. The concluding sentence of the discussion already is more appropriate as that indeed only mentions one target (being FABP3), rather than an range of targets/pathways, which are unlikely to be selectively hit by one single therapeutic intervention.

We agree that the sentence in the Abstract conclusion was overstated. As requested by the reviewer, we have revised it as follows: "Therefore, FABP3 drives membrane lipid composition-mediated ER stress to regulate muscle homeostasis during aging and is a valuable target for sarcopenia." (page 2, line 13)

- Validity:

The main flaws of the manuscript are an incomplete and hence biased assessment of the potential signaling pathways that could mediate the functional outcomes along with incomplete mechanistic support for the stated conclusions.

- Originality and significance:

The identity of FABP3 as a possible target for age associated sarcopenia is novel and interesting and warrants further investigation.

- Data & methodology: No major issues here

- Appropriate use of statistics and treatment of uncertainties: No major issues here

- Conclusions: Do you find that the conclusions and data interpretation are robust, valid and reliable?

Although the main conclusions are reasonably well supported, the authors have overstated some of the possible mechanistic links.

- Suggested improvements: Please list additional experiments or data that could help strengthening the work in a revision.

See in the section 'To the authors' .

- References: Does this manuscript reference previous literature appropriately? If not, what references should be included or excluded?

No problems here

- Clarity and context: Is the abstract clear, accessible? Are abstract, introduction and conclusions appropriate?

No major issues here, except for the strong link that is made that 'FABP3-lipid composition-membrane fluidity-ER stress axis is a target for sarcopenia treatment'. Basically this is not a single target but rather quite a range of diverse mechanisms which indeed all are likely to affect sarcopenia, but which can be hardly viewed as a single target for treatment.

References

Afroze, D. and A. Kumar (2019). "ER stress in skeletal muscle remodeling and myopathies." FEBS J **286**(2): 379-398.

Alberts, B., A. Johnson and J. Lewis (2002). The art of MBoC4: the complete set of figures from Molecular biology of the cell 4th edition. London, Garland.

Bourbon, N. A., J. Yun and M. Kester (2000). "Ceramide directly activates protein kinase C zeta to regulate a stress-activated protein kinase signaling complex." J Biol Chem **275**(45): 35617-35623.

Cao, H., M. Sekiya, M. E. Ertunc, M. F. Burak, J. R. Mayers, A. White, K. Inouye, L. M. Rickey, B. C. Ercal, M. Furuhashi, G. Tuncman and G. S. Hotamisligil (2013). "Adipocyte lipid chaperone AP2 is a secreted adipokine regulating hepatic glucose production." Cell Metab **17**(5): 768-778.

Caron, A. Z., G. Drouin, J. Desrosiers, F. Trenz and G. Grenier (2009). "A novel hindlimb immobilization procedure for studying skeletal muscle atrophy and recovery in mouse." J Appl Physiol (1985) **106**(6): 2049-2059.

Childs, T. E., E. E. Spangenburg, D. R. Vyas and F. W. Booth (2003). "Temporal alterations in protein signaling cascades during recovery from muscle atrophy." Am J Physiol Cell Physiol **285**(2): C391-398.

Degens, H. and S. E. Alway (2003). "Skeletal muscle function and hypertrophy are diminished in old age." Muscle Nerve **27**(3): 339-347.

Erbay, E., V. R. Babaev, J. R. Mayers, L. Makowski, K. N. Charles, M. E. Snitow, S. Fazio, M. M. Wiest, S. M. Watkins, M. F. Linton and G. S. Hotamisligil (2009). "Reducing endoplasmic reticulum stress through a macrophage lipid chaperone alleviates atherosclerosis." Nat Med **15**(12): 1383-1391.

Fan, W. and R. M. Evans (2015). "Turning up the heat on membrane fluidity." Cell **161**(5): 962-963.

Furuhashi, M. and G. S. Hotamisligil (2008). "Fatty acid-binding proteins: role in metabolic diseases and potential as drug targets." Nat Rev Drug Discov **7**(6): 489-503.

Gomes, A. P., N. L. Price, A. J. Ling, J. J. Moslehi, M. K. Montgomery, L. Rajman, J. P. White, J. S. Teodoro, C. D. Wrann, B. P. Hubbard, E. M. Mercken, C. M. Palmeira, R. de Cabo, A. P. Rolo, N. Turner, E. L. Bell and D. A. Sinclair (2013). "Declining NAD(+) induces a pseudohypoxic state disrupting nuclear-mitochondrial communication during aging." Cell **155**(7): 1624-1638.

Kim, J. H., P. Jin, R. Duan and E. H. Chen (2015). "Mechanisms of myoblast fusion during muscle development." Curr Opin Genet Dev **32**: 162-170.

Krawiec, B. J., R. A. Frost, T. C. Vary, L. S. Jefferson and C. H. Lang (2005). "Hindlimb casting decreases muscle mass in part by proteasome-dependent proteolysis but independent of protein synthesis." Am J Physiol Endocrinol Metab **289**(6): E969-980.

Krishnamoorthy, T., G. D. Pavitt, F. Zhang, T. E. Dever and A. G. Hinnebusch (2001). "Tight binding of the phosphorylated alpha subunit of initiation factor 2 (eIF2alpha) to the regulatory subunits of guanine nucleotide exchange factor eIF2B is required for inhibition of translation initiation." Mol Cell Biol **21**(15): 5018-5030.

Miyake, M., M. Kuroda, H. Kiyonari, K. Takehana, S. Hisanaga, M. Morimoto, J. Zhang, M. Oyadomari, H. Sakaue and S. Oyadomari (2017). "Ligand-induced rapid skeletal muscle atrophy in HSA-Fv2E-PERK transgenic mice." PLoS One **12**(6): e0179955.

Park, S. K. and T. A. Prolla (2005). "Gene expression profiling studies of aging in cardiac and skeletal muscles." Cardiovasc Res **66**(2): 205-212.

Pelsters, M. M., W. T. Hermens and J. F. Glatz (2005). "Fatty acid-binding proteins as plasma markers of tissue injury." Clin Chim Acta **352**(1-2): 15-35.

Pritt, M. L., D. G. Hall, J. Recknor, K. M. Credille, D. D. Brown, N. P. Yumibe, A. E. Schultze and D. E. Watson (2008). "Fabp3 as a biomarker of skeletal muscle toxicity in the rat: comparison with conventional biomarkers." Toxicol Sci **103**(2): 382-396.

Richieri, G. V., R. T. Ogata and A. M. Kleinfeld (1994). "Equilibrium constants for the binding of fatty acids with fatty acid-binding proteins from adipocyte, intestine, heart, and liver measured with the fluorescent probe ADIFAB." J Biol Chem **269**(39): 23918-23930.

Siu, P. M. and S. E. Alway (2005). "Age-related apoptotic responses to stretch-induced hypertrophy in quail slow-tonic skeletal muscle." Am J Physiol Cell Physiol **289**(5): C1105-1113.

Slimani, L., E. Vazeille, C. Deval, B. Meunier, C. Polge, D. Dardevet, D. Bechet, D. Taillandier, D. Micol, A. Listrat, D. Attaix and L. Combaret (2015). "The delayed recovery of the remobilized rat tibialis anterior muscle reflects a defect in proliferative and terminal differentiation that impairs early regenerative processes." J Cachexia Sarcopenia Muscle **6**(1): 73-83.

Suetta, C., C. Clemmensen, J. L. Andersen, S. P. Magnusson, P. Schjerling and M. Kjaer (2010). "Coordinated increase in skeletal muscle fiber area and expression of IGF-I with resistance exercise in elderly post-operative patients." Growth Horm IGF Res **20**(2): 134-140.

Vance, D. E. and J. E. Vance (2008). Biochemistry of lipids, lipoproteins and membranes (5th Edn.). Amsterdam; London, Elsevier.

White, J. R., A. L. Confides, S. Moore-Reed, J. M. Hoch and E. E. Dupont-Versteegden (2015). "Regrowth after skeletal muscle atrophy is impaired in aged rats, despite similar responses in signaling pathways." Exp Gerontol **64**: 17-32.

Reviewers' Comments:

Reviewer #1:

Remarks to the Author:

In the revised manuscript 'FABP3-mediated membrane lipid saturation alters fluidity and induces ER stress in skeletal muscle aging', the authors have performed all the experiments that were suggested. They have addressed all the concerns satisfactorily.